# Swarming bacteria undergo localized dynamic phase transition to form stress-induced biofilms

Iago Grobas[1], Marco Polin[1,2,3,4†]*, Munehiro Asally[5,6,7†]*

[1]Warwick Medical School, Universityof Warwick, Coventry, United Kingdom; [2]Centre for Mechanochemical Cell Biology, University of Warwick, Coventry, United Kingdom; [3]Physics Department, University of Warwick, Coventry, United Kingdom; [4]Mediterranean Institute for Advanced Studies (IMEDEA UIB-CSIC), C/ Miquel Marqués, Balearic Islands, Spain; [5]Bio-Electrical Engineering Innovation Hub, University of Warwick, Coventry, United Kingdom; [6]Warwick Integrative Synthetic Biology Centre, University of Warwick, Coventry, United Kingdom; [7]School of Life Sciences, University of Warwick, Coventry, United Kingdom

**Abstract** Self-organized multicellular behaviors enable cells to adapt and tolerate stressors to a greater degree than isolated cells. However, whether and how cellular communities alter their collective behaviors adaptively upon exposure to stress is largely unclear. Here, we investigate this question using *Bacillus subtilis*, a model system for bacterial multicellularity. We discover that, upon exposure to a spatial gradient of kanamycin, swarming bacteria activate matrix genes and transit to biofilms. The initial stage of this transition is underpinned by a stress-induced multilayer formation, emerging from a biophysical mechanism reminiscent of motility-induced phase separation (MIPS). The physical nature of the process suggests that stressors which suppress the expansion of swarms would induce biofilm formation. Indeed, a simple physical barrier also induces a swarm-to-biofilm transition. Based on the gained insight, we propose a strategy of antibiotic treatment to inhibit the transition from swarms to biofilms by targeting the localized phase transition.

*For correspondence:
M.Polin@warwick.ac.uk (MP);
m.asally@warwick.ac.uk (MA)

†These authors contributed equally to this work

Competing interests: The authors declare that no competing interests exist.

## Introduction

The ability to sense, respond, and adapt to varieties of chemical, physical, and environmental stresses is fundamental to the survival of organisms. In addition to general stress-response pathways at individual cell level, which activate target genes in response to a variety of stresses, multicellular systems can tolerate stresses through self-organization, the emergence of order in space and time resulting from local interactions between individual cells (*Ben-Jacob et al., 2000*; *Kirschner et al., 2000*; *Schweisguth and Corson, 2019*). Bacterial biofilm formation and swarming are ancient forms of multicellular adaptation (*de la Fuente-Núñez et al., 2013*; *Lyons and Kolter, 2015*), where cells can coordinate their behaviors through chemical (*Daniels et al., 2004*; *Xavier, 2011*), mechanical (*Grobas et al., 2020*; *Be'er and Ariel, 2019*; *Mazza, 2016*), and bioelectrical (*Benarroch and Asally, 2020*; *Prindle et al., 2015*) interactions. Biofilm cells are much more tolerant to various stresses than genetically identical cells in isolation (*Meredith et al., 2015*), owing to the physico-chemical properties of extracellular polysubstance (EPS) (*Flemming and Wingender, 2010*), meta-bolic coordination (*Liu et al., 2015*), slow cell growth (*Costerton et al., 1999*), and in the case of air-exposed biofilms, diffusion barrier by the archetypical wrinkled morphology (*Epstein et al., 2011*; *Vlamakis et al., 2013*).

Swarming bacteria can also collectively tolerate antibiotic treatments that are lethal to planktonic cells – albeit only to a lesser degree than biofilms – through motility-induced mixing and reduced

small-molecule absorption (*Bhattacharyya et al., 2020*; *Butler et al., 2010*; *Lai et al., 2009*). As a rapid mode of surface colonization (*Kearns, 2010*), a swarm's ability to withstand high antibiotic concentrations could therefore lead to the subsequent establishment of highly resilient biofilms in regions that could not have been reached otherwise. The formation of biofilms is linked to general stress response pathways (*Lories et al., 2020*; *Nadezhdin et al., 2020*); it has been reported that biofilms can be induced from free-swimming planktonic cells by a wide range of biochemical and mechanical stressors, such as aminoglycoside antibiotics (*Hoffman et al., 2005*; *Jones et al., 2013*), redox-active compounds (*Wang et al., 2011*), nutrient depletion (*Zhang et al., 2014*), and mechanical stress (*Chu et al., 2018*). However, whether swarming collectives may also transit into more resilient biofilms upon exposure to stressors, and -if so- how such a transition can be initiated within a self-organized swarm, are unknown.

Biophysics of collective motion in bacteria, such as flocking and swarming, is a major topic within the rapidly growing research area of active matter (*Bechinger et al., 2016*; *Geyer et al., 2019*; *Vicsek et al., 1995*). Theoretical models of active matter have repeatedly predicted that at a sufficient concentration, a collection of motile particles, be it swimming cells or self-propelled artificial microswimmers, can spontaneously form high-density clusters (*Barré et al., 2015*; *Gonnella et al., 2015*). Indeed, this process has been implicated in the development of cell inhomogeneities leading to fruiting-body formation in *M. xanthus* (*Liu et al., 2019*). This transition, known as Motility Induced Phase Separation (MIPS), is based on feedback between the localized decrease in particles' speed at high concentration, caused by physical interactions, and the spontaneous accumulation of active particles in the clusters where their speed is lower (*Gonnella et al., 2015*). When the particles' speed is sufficiently high and their concentration is in the appropriate range (typical volume fractions of 0.3–0.8 or 0.6–0.7 for round or rod-shaped particles, respectively [*Digregorio et al., 2018*; *van Damme et al., 2019*]), inherent density fluctuations are amplified by the particles' slowing down, and the system effectively phase separates into high-density/low-motility clusters surrounded by a low-density/high-motility phase (*Cates and Tailleur, 2015*).

The theory of MIPS, then, suggests that persistent heterogeneity in cell density – the MIPS clusters – can develop spontaneously when both cell speed and density are appropriate (*Figure 1*, gray U-shape region) (*Cates et al., 2010*). Such conditions should be achievable within a bacterial swarm. Given that cell-density heterogeneity can lead to the production of matrix and biofilm formation mediated by localized cell death (*Asally et al., 2012*; *Ghosh et al., 2013*), we hypothesized that the heterogeneity caused by putative MIPS-like clusters could in turn underpin a transition from bacterial swarms into biofilms (*Figure 1*). As MIPS-like clustering is an emergent phenomenon arising from physical interactions between individual agents, it may endow swarms with a collective response to a wide spectrum of stressors that cause changes in cell motility and/or density.

Here, we show that *Bacillus subtilis* swarms can indeed transit into biofilms through a MIPS-like process, induced by physical or chemical stresses applied at the swarming front. When swarming cells are exposed to kanamycin, they activate the expression of the biofilm matrix operon, *tapA-sipW-tasA*, and eventually develop wrinkled biofilms. The transition is initiated by a localized phase transition, where the expanding swarming monolayer generates multilayer clusters of cells as a consequence of motility- and stress-induced elevation of cell density. Based on the insights gained from our investigation, we show that targeting the multilayered region by administering a given amount of antibiotic in two separate doses is effective in inhibiting the formation of wrinkly biofilms from swarming cells.

## Results

### Swarming *B. subtilis* transits into a biofilm in presence of a spatial kanamycin gradient

To examine if a stressor triggers biofilm formation from a swarming collective, we performed swarming assays using *B. subtilis* with a spatial gradient of the aminoglycoside kanamycin. Specifically, a disk containing 30 µg of kanamycin was placed on the side of a swarming plate (0.5% agar) and allowed to rest for 24 hr to establish a space-dependent antibiotic concentration profile (*Figure 2a* and *Figure 2—figure supplement 1*). An inoculum of *B. subtilis* culture was then placed at the center of the dish, ~4 cm away from the kanamycin source, and the plate was imaged while being kept

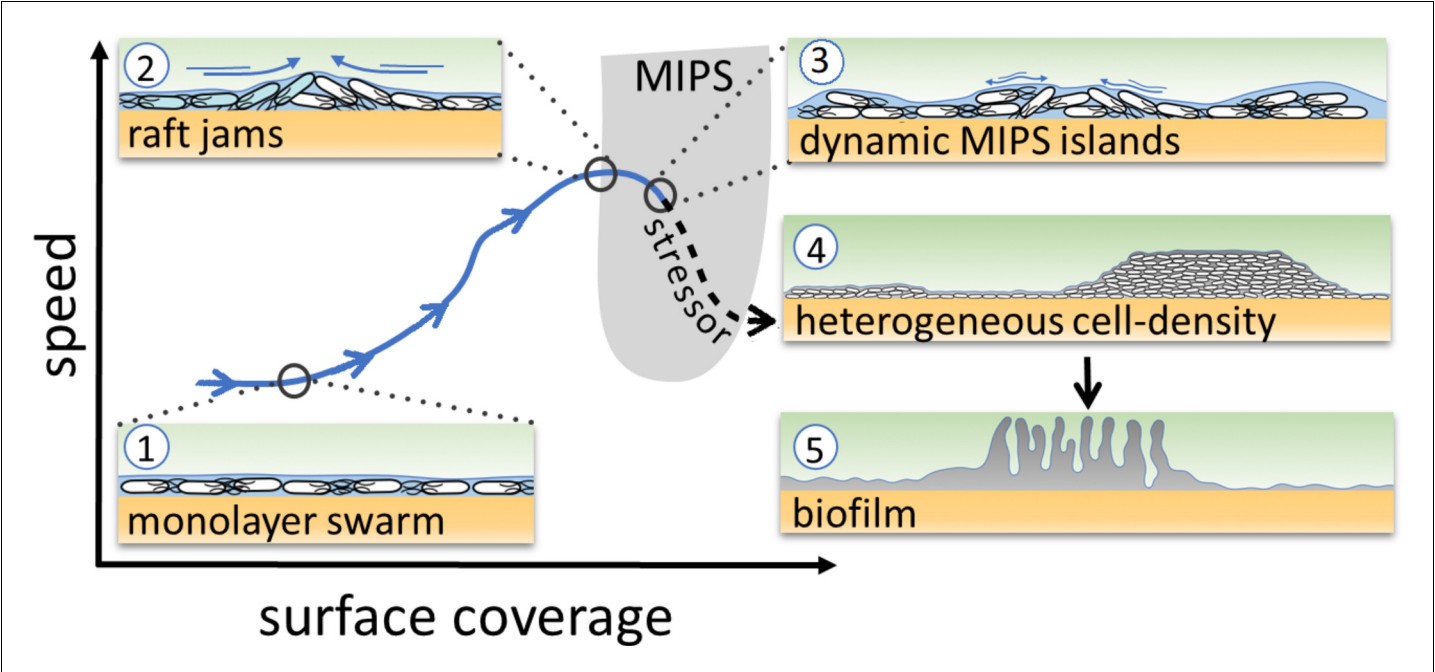

**Figure 1.** Schematic of the transition from swarming to biofilm formation through MIPS. *B. subtilis* cells swarm in a monolayer (stage 1). As cell density increases, cells form fast-moving rafts which can collide and form transient jams. Cells at the boundary between the colliding rafts are pushed upwards and protrude from the surrounding monolayer (stage 2). Further increase in surface coverage can promote the formation of dynamic MIPS-like islands where cells accumulate within the swarm while still being dynamic (stage 3). Eventually, this uneven distribution of cells gives rise to macroscopic spatial heterogeneity in cell density (stage 4), which can lead to the formation of biofilms (stage 5).

at 30°C (*Figure 2a*). After ~2 hr of lag period, the cells formed a rapidly expanding swarming front (~4 mm/hr), which became progressively slower towards the source of kanamycin until it completely stopped ~0.3 cm away from the disk (*Figure 2—figure supplement 2* and *Video 1*). After further incubation at 30°C for 36 hr, the colony developed prominent wrinkles, the morphological hallmark of *B. subtilis* pellicles and colony biofilms (*Cairns et al., 2014*; *Vlamakis et al., 2013*), across a ~ 3 mm band ~1.3 cm away from the kanamycin disk (*Figure 2b*). The estimated level of kanamycin at the site was around the minimum inhibitory concentration (MIC) of planktonic cells (*Figure 2—figure supplement 1*).

To quantify the degree of biofilm formation, we measured the characteristic wavelength and roughness of the wrinkles. In this study, the term biofilm is interpreted as wrinkly biofilm, and the wrinkles' wavelength is used as a convenient indirect quantification of biofilm formation since the wavelength has been reported to correlate with biofilm stiffness and extracellular matrix (*Asally et al., 2012*; *Kesel et al., 2016*; *Yan et al., 2019*). More specifically, the wavelength is smaller in colonies of matrix mutants and greater in hyper biofilm-forming mutants. The wrinkles appearing near the kanamycin disk had a wavelength of $\lambda = 560$ µm and roughness $r = 10$, indicating a much stiffer biofilm than the case without kanamycin, which only produced a faint small-wavelength surface roughness ($\lambda = 91$ µm, $r = 5.5$; *Figure 2—figure supplement 3*).

To further verify the association of these wrinkles to biofilms, we repeated the assay using Δ*eps*, a mutant known to be impaired in biofilm formation (*Nagorska et al., 2010*; *Branda et al., 2001*) but capable of swarming (*Video 2*), and confirmed no wrinkles with this mutant (*Figure 2c*). Furthermore, we measured the expression of *tasA*, which encodes an essential matrix component for *B. subtilis* biofilm formation (*Romero et al., 2010*), using a strain carrying P$_{tapA}$-*yfp*, a fluorescence reporter for the expression of the *tapA-sipW-tasA* operon (*Vlamakis et al., 2008*). The result showed a greater fluorescence signal from the promoter reporter when swarming cells were exposed to kanamycin (*Figure 2e*), suggesting that the exposure to kanamycin results in higher abundance in extracellular matrix and induces biofilm formation in a swarming colony on a soft-agar plate.

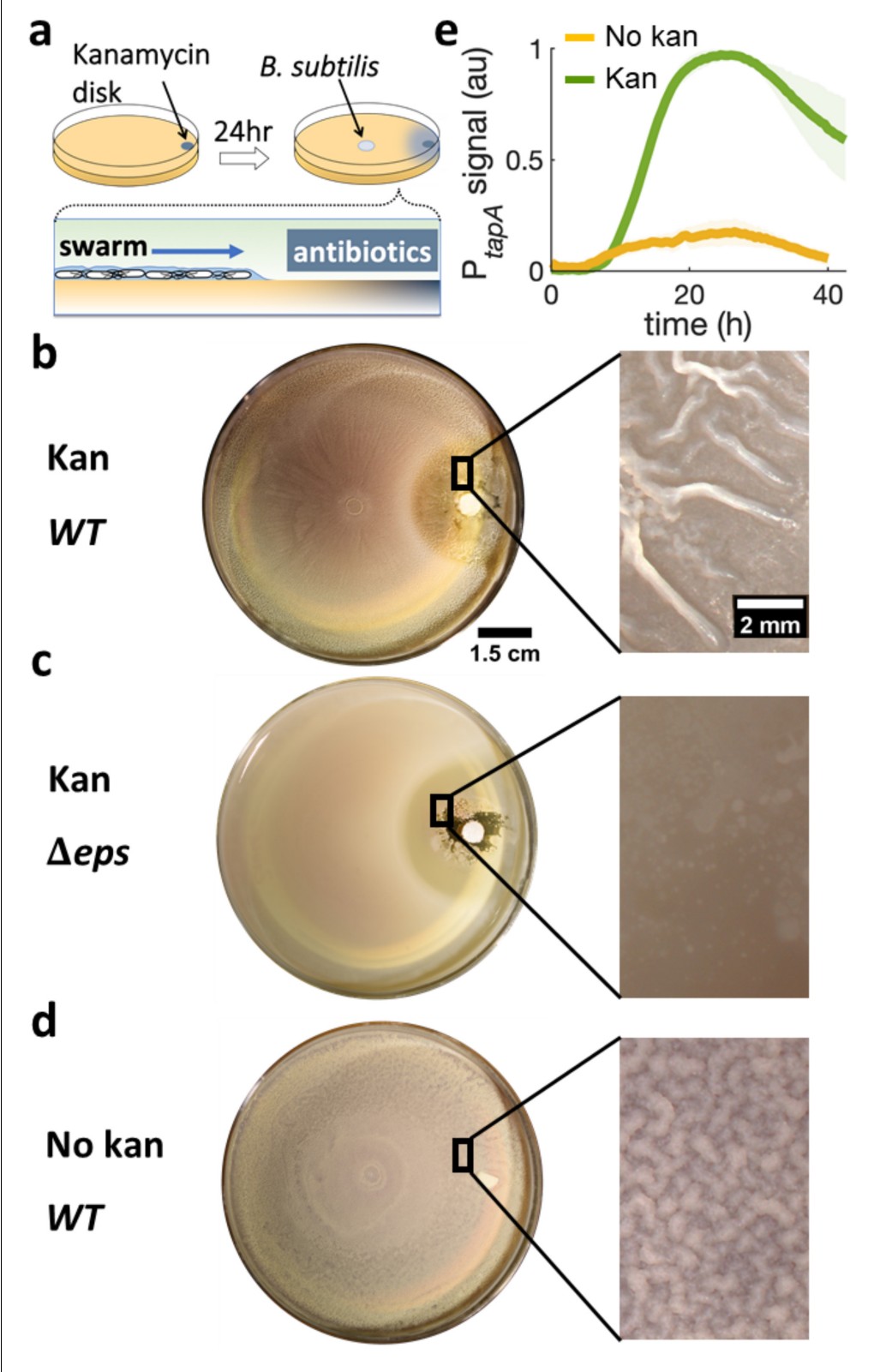

**Figure 2.** Swarming cells transit into biofilm in the presence of kanamycin gradient. **(a)** Schematics of swarming bacteria expanding from the center of a 9-cm Petri dish toward a kanamycin-diffusive disk. Kanamycin disk was placed for 24 hr to form a spatial gradient (*Figure 2—figure supplement 1*). **(b, c, d)** Swarming *B. subtilis* plates after 40 hr incubation. Wrinkles are formed at the region ~2 cm away from disk with **(b)** wildtype (WT), but not with **(c)** Δeps deletion strain. **(d)** WT swarming plate with a diffusive disk without antibiotics. Zoomed images show the colony surface. **(e)** Mean fluorescence

*Figure 2 continued on next page*

*Figure 2 continued*

intensity of P$_{tapA}$-*yfp* was measured in the region just after the depletion zone created by the kanamycin-diffusive disk (green). A region in a similar location was measured in absence of kanamycin (yellow). The mean was taken from three independent experiments. The shady areas represent the s.e.m.

The online version of this article includes the following figure supplement(s) for figure 2:

**Figure supplement 1.** Evolution of the kanamycin profile along the plate measured from the kanamycin disk.
**Figure supplement 2.** Speed of the swarming front when approaching the diffusive disk.
**Figure supplement 3.** Quantification of wrinkle formation for different regions in the kanamycin plate and in the blank case.
**Figure supplement 4.** Distance where the wrinkles appear depending on the amount of kanamycin in the diffusive disk.
**Figure supplement 5.** The plate inoculated with swarming *B. subtilis* showed no wrinkles even after a week incubation at 30℃.
**Figure supplement 6.** Four μl of *B. subtilis* were spotted in four different points on a hard agar plate.
**Figure supplement 7.** Comparison of wrinkle formation between hard agar and swarming agar in presence of a gradient of kanamycin.

The emergence of wrinkles across a ~ 3 mm-band away from kanamycin suggests that the swarm-to-biofilm transition corresponds to exposure to a certain concentration range of kanamycin. Indeed, increasing the initial concentration of antibiotic in the disk, wrinkles emerged further away from the disk (*Figure 2—figure supplement 4*), while a disk without antibiotic did not promote wrinkle formation (*Figure 2d* and *Figure 2—figure supplement 5*). Intriguingly, spotting individual colonies on hard agar plates (1.5%) typically used for biofilm colony assays, showed that kanamycin inhibited biofilm formation (*Figure 2—figure supplement 6*). When cells were instead uniformly spread across the hard agar plate at concentrations similar to those of a swarming monolayer, the system developed only faint wrinkles around the antibiotic exclusion zone (*Figure 2—figure supplement 7*). The fact that a kanamycin gradient promotes wrinkle formation significantly more in a swarming colony than a non-motile culture, pointed to a fundamental role played by cell motility and suggested the need to investigate the swarming dynamics in detail.

## The emergence of the biofilm is templated by a localized transition from mono- to multilayers at the swarming front

To investigate the potential link between swarming dynamics and biofilm formation, we characterized the swarming dynamics at the single-cell level, with and without a kanamycin gradient. We combined time-lapse imaging at both microscopic (10×; 30 fps) and macroscopic (2×; 0.006 fps) scales to capture both swarming, which is microscopic and fast (~70 μm² cell rafts; speed ~60 μm/s), and biofilm formation, which occurs at a macroscopic scale over hours to days. For the microscopic imaging, we focused on the cells at ~1 cm from the kanamycin disk, where wrinkles eventually appear and kanamycin level is around MIC for planktonic cells (*Figure 2—figure supplement 1*). The cells displayed typical swarming dynamics during the expansion of the colony front (*Figure 3a*). As time progressed, we observed the local surface coverage of the monolayer swarm to progressively increase from initial values of ~20% to ≥60%, at which point the swarming rafts started displaying jamming events lasting ~1–2 s, during which cell speed was strongly reduced and groups of cells protruded temporarily from the swarming monolayer (typical size of jammed

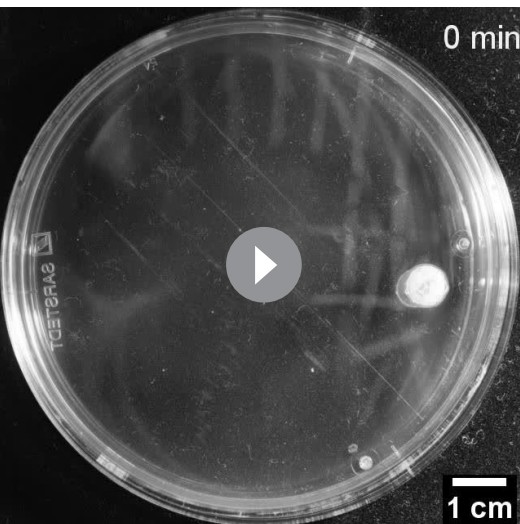

**Video 1.** Expansion of swarming bacteria in presence of a gradient of kanamycin. Wild-type *Bacillus subtilis* was inoculated at the center of the plate. Eventually, bacteria at the swarming front halt due to the presence of the kanamycin-diffusive disk placed on a side of a 9-cm Petri Dish 24 hr prior to inoculation. After the front stops, the swarm is still dynamic since multiple waves of swarming bacteria come from the center of the plate.
https://elifesciences.org/articles/62632#video1

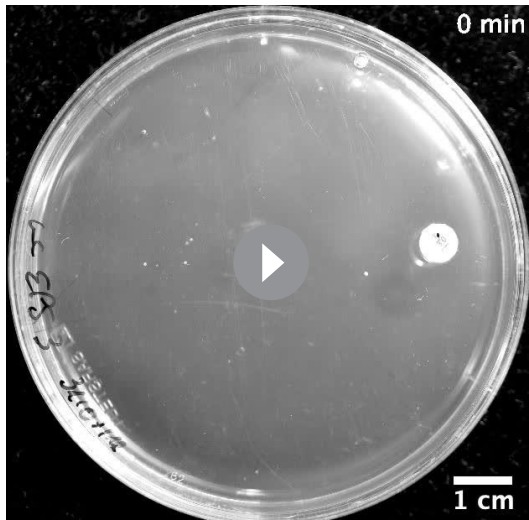

**Video 2.** Swarming *Bacillus subtilis* Δeps deletion strain was inoculated at the center of the plate. Eventually, bacteria at the swarming front halt due to the presence of the kanamycin diffusive disk placed on a side of a 9 cm Petri dish 24 hr before inoculation.
https://elifesciences.org/articles/62632#video2

group $\sim$500 $\mu m^2$, see *Video 3*). Interestingly, in this area we did not observe a significant fraction of immotile cells, as previously reported for swarming in presence of a uniform concentration of kanamycin (*Benisty et al., 2015*).

The jamming of aligned rafts has been predicted numerically for elongated self-propelled particles on a plane (*Peruani et al., 2011*; *Weitz et al., 2015*), although in that case the planar confinement precluded any possible excursion in the third dimension. With a slight further increase in local cell density, the temporary jams led to permanent isolated multilayered regions, which we call 'islands', typically $\sim$27,000 $\mu m^2$ in size and with constantly fluctuating boundaries (*Figure 3b* inset, *Video 4*). The islands did not appear to be related to any evident inhomogeneity of the underlying agar. Consistent with a previous study reporting the reduction in speed by kanamycin (*Benisty et al., 2015*), we observed the speed declined before island formation (*Figure 3—figure supplement 1*). Within islands, cells were highly dynamic and appeared to meander across the layers albeit at a reduced speed (*Figure 3—figure supplement 2*, *Video 5*). Cell movement within islands high-lights the fact that these are formed by actively motile cells, rather than groups of immotile bacteria which spontaneously separate from the monolayer. In particular, we did not find any evidence for clusters of immotile cells templating the islands. Rather, motile cells were constantly exchanged between the multilayer and the surrounding monolayer. Following a continuous increase in cell density with time, islands grew in size and merged with each other, eventually forming a much larger multilayered region (*Figure 3c*, *Video 6*). The nucleation and merging of islands were observed up to four layers before we lost the ability to recognize further transitions due to a flattening of the image contrast (*Figure 3—figure supplement 3*).

The formation of islands was also observed by macroscopic time-lapse imaging. In absence of kanamycin, islands emerged simultaneously throughout the plate, forming an intricate granular pattern with a typical size of $\sim$2000 $\mu m^2$ (*Figure 3d*), a phenomenology reminiscent of spinodal decomposition in binary fluids (*Qiu et al., 2001*). In contrast, the process of islands formation and growth was markedly different in presence of kanamycin (*Figure 3e*). After the swarming front expansion was stopped due to inhibition by the antibiotic, well-defined, distinct islands first emerged within a $\sim$ 5 mm-wide band just inside of the arrested swarming front, and then grew with a strongly aniso-tropic pattern oriented transversally to the front of the swarm (*Video 6*). This phenomenology is reminiscent of a binodal – rather than spinodal – liquid-liquid phase separation in a temperature gradient (*Bartolini et al., 2019*) which here might reflect a kanamycin-induced speed gradient. The resulting macroscopic cell-density heterogeneity was followed by the subsequent formation of biofilm wrinkles, $\sim$1 cm away from the disk (*Figure 2b*).

Since quorum sensing is often associated with bacterial collective behavior, we wondered if quorum sensing may play a role in the emergence of islands and the ensuing multilayer. We therefore repeated the experiments with Δ*phrC* and Δ*opp* mutant strains, lacking the Phr quorum-sensing system in *B. subtilis* but still capable of swarming. In both cases, we observed the wild-type phenomenology of wrinkled biofilms and the emergence of islands (*Figure 3—figure supplement 4*). The results indicated that this quorum-sensing system is not responsible for the multilayer transition, suggesting that exposure to kanamycin can promote cell-density heterogeneity by multilayer islands formation in a quorum-sensing independent manner.

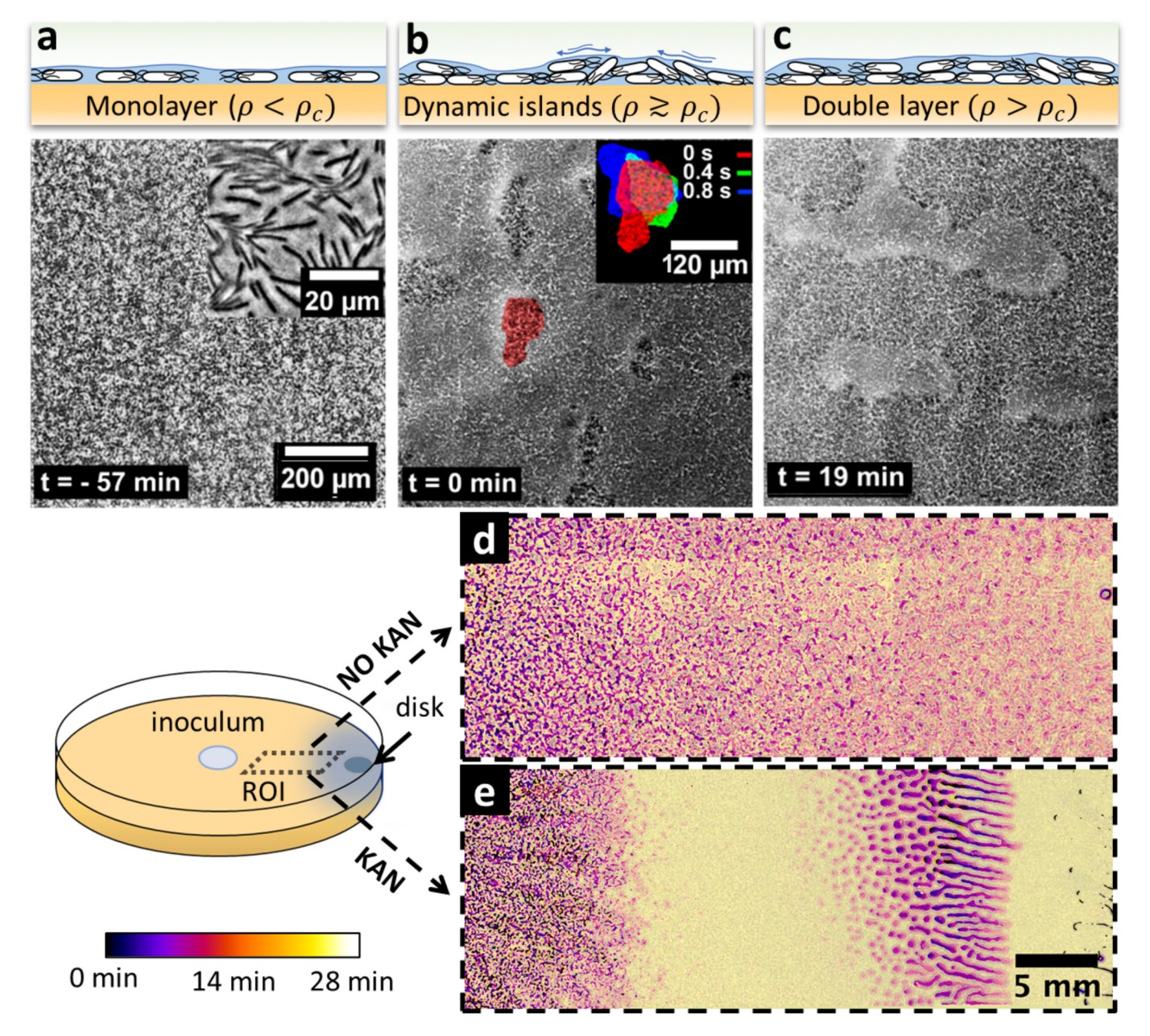

**Figure 3.** Swarming bacteria form patterned multilayer regions in presence of kanamycin. (a, b, c) Microscopy images of swarming bacteria and side-view illustrations at different levels of cell density ($\rho$). Timestamp is relative to the formation of dynamic multilayer islands. (a) When cell density is below a threshold $\rho_c$, cells swarm in a monolayer. Image is 57 min prior to islands formation. Inset is a zoomed image showing swarming rafts. (b) Increase in cell density leads to formation of second layers, that appear as darker regions in the swarm. These islands are highly dynamic, and cells are motile. The shape of the islands changes dynamically within 1 s (insert). See also *Video 4*. (c) Over time, the islands increase their size and merge together to form double-layered regions, coexisting with mono-layered regions. Double-layered regions are seen darker than mono layered regions. (d, e) Images of swarming colony with and without kanamycin. The diagram illustrates the regions of interest (ROI) used for the panels. The dynamics of island formation are color coded using the lookup table. The origin of times is the appearance of island. The earliest islands correspond to the darkest colors whereas the new features correspond to the lightest ones. See also *Video 6*. (d) In the absence of kanamycin, multilayer regions are much grainer with no clear patterns of propagation. (e) In the presence of kanamycin, multilayered regions have a defined pattern, starting from the regions closer to kanamycin to form an elongated shape. These regions appear predominantly at ~7 mm away from kanamycin.

The online version of this article includes the following figure supplement(s) for figure 3:

**Figure supplement 1.** Cell speed at 0.7, 1.7, and 2.7 cm away from the kanamycin disk for a time interval of 1 hr before the initial formation of the islands (0 min corresponds to island formation at 1.7 cm).

*Figure 3 continued on next page*

**Figure supplement 2.** Average speed in the monolayer vs speed within the island.
**Figure supplement 3.** Asymmetric multilayer formation along time in presence of kanamycin.
**Figure supplement 4.** Islands and posterior wrinkle formation in presence of a kanamycin gradient by the quorum sensing knockout strains Δ*phrc* and Δopp.

## Transition from monolayer to multilayer resembles motility induced phase separation

To address the mechanism of multilayer formation, we wondered if physical stresses may be responsible for this emergent collective behavior since single-to-multilayer transitions have been reported for confined bacterial aggregates, either growing or gliding, as a consequence of build-up of internal mechanical stresses (*Takatori and Mandadapu, 2003*; *Grant et al., 2014*; *Su et al., 2012*). In particular, we considered if the active matter model of MIPS for rod-shape particles (*Cates and Tailleur, 2015*) could act as a useful paradigm for understanding the emergence of multilayer regions. We therefore mapped our experimental results onto the typical phase space considered when studying MIPS transitions, where an active two-dimensional system is characterized by its surface coverage, $\varphi$, and its rotational Péclet number, $Pe_r = u/LD_r$. The latter is defined in terms of the average speed u, characteristic size L, and rotational diffusivity $D_r$ of the active particles. MIPS clusters are expected within a U-shaped region characterized by a sufficiently large $Pe_r$ and a range of surface coverages that, for rod-like particles like *B. subtilis*, is pushed to values higher than the ~50%, typical of circular particles, due to antagonistic effects of cell-to-cell alignment (*Figure 4a*, gray U-shape region shows the prediction for aspect ratio two from *van Damme et al., 2019*). We then quantified the cell density (surface coverage, $\varphi$) and the rotational Péclet number from the different stages of the swarming process. *Figure 4a* shows these two quantities for cells in the monolayer (blue dots) and for bacterial jams (red dots). While bacteria in the monolayer corresponded overwhelmingly to points well outside the MIPS region (*Figure 4a* blue dots), bacterial jams – the first stage in the development of stable multilayer islands – clustered around the area predicted for MIPS in two-dimensional self-propelled disks (*Figure 4*, red dots; MIPS region from *van Damme et al., 2019*). Jamming events were also characterized by a sudden drop in cells' speed (*Figure 4b* and *Video 3*), consistent with the basic premise of MIPS. This is only a temporary slowing down due to cell-cell interactions rather than the persistent decrease in motility that can be induced by prolonged exposure to kanamycin (*Benisty et al., 2015*). Altogether, these results show that the mechanism leading to the development of multilayered islands is compatible with the MIPS process in active matter. Having established the similarities between MIPS and multilayer formation in the swarm, it is important to test the ability of the model to predict the results of new experiments.

## Local accumulation of swarming cells induces multilayer transition and biofilm formation

The MIPS paradigm makes the experimentally testable prediction that it should be possible to induce multilayer formation by altering the local density of motile cells, thereby forcing the system to enter the MIPS region in phase space (*Figure 4a*, shaded region). We tested this in two ways: by UV irradiation, and through the use of a physical barrier to block front expansion. Near-UV light can decrease cell speed in gram-negative bacteria like *E. coli* and *S. marcescens* (*Krasnopeeva et al., 2019*; *Yang et al., 2019*) as well as, as we report here, in *B. subtilis* (*Figure 4—figure supplement 1*). Locally slowing down motility by light can lead to a local

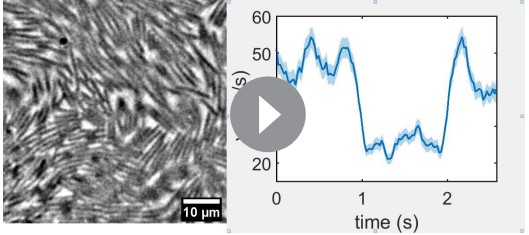

**Video 3.** Swarming bacteria during the jamming process visualized as darker cells in the field of view observed at ×60 magnification. The speed plot shows the average speed in the field of view and the blue shade represents s.e.m. The vertical line in the graph indicates the frame of the time-lapse in which the average speed was calculated. It can be observed that the jam makes the overall speed to drop by half of the initial magnitude.

https://elifesciences.org/articles/62632#video3

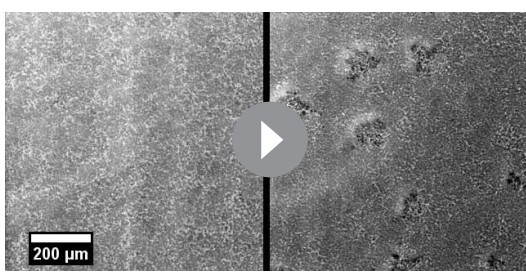

**Video 4.** Process of jams and islands formation under ×10 magnification. The left panel shows swarming bacteria continuously jamming. The jams look like dark localized small regions in the field of view which continuously appear and disappear. The right panel shows the same scenario 10 min later, when the islands become more stable. Islands look like extended dark regions in the field of view which are highly dynamic and with highly fluctuating boundaries.
https://elifesciences.org/articles/62632#video4

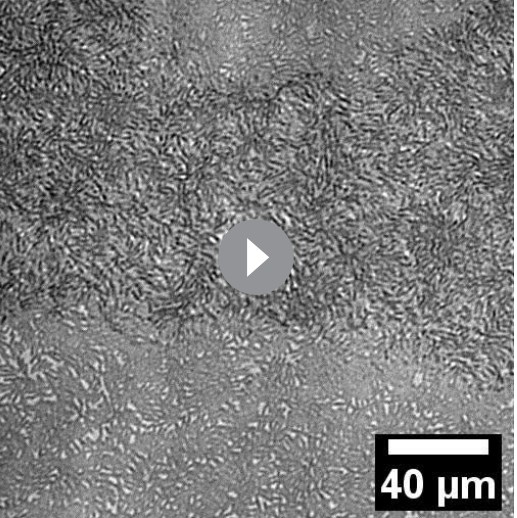

**Video 5.** Coexistence between a second layer of swarming bacteria (darker region in the field of view) and a surrounding monolayer of swarming cells under ×40 magnification. The boundaries of the island continuously exchange cells with the monolayer surrounding it. Within the island, cell appear to meander across the layers.
https://elifesciences.org/articles/62632#video5

increase in cell concentration through the accumulation of cells from non-irradiated regions (*Figure 4c*). We first verified that the illumination by UV light itself does not induce a multilayer transition, by illuminating an area >200 fold greater than the field of view. This prevented cells outside of the irradiated area from accumulating within the field of view during the experiment. Accordingly, the surface coverage increased only marginally ($\varphi \simeq 0.45$ to $\varphi \simeq 0.47$ in 3 min; *Figure 4—figure supplement 1*). The UV illumination caused an initial sudden drop in average speed from 65 to 50 µm/s (*Figure 4—figure supplement 1*), followed by a nonlinear progressive slow down over the course of 3 min. Within the phase space picture (*Figure 4a*, *Figure 4—figure supplement 1b*), this corresponds to a trajectory that essentially just moves towards progressively lower values of $Pe_r$. This should not lead to either jams or island formation, which in fact were never observed (*Figure 4a*, *Figure 4—figure supplement 1b*).

We next illuminated a region of size similar to the field of view. This arrangement allowed cells from the outer region to accumulate within the field of view due to UV-induced slow-down, a phenomenon which is a direct consequence of the active, out-of-equilibrium nature of the swarm with no counterpart in statistical systems in equilibrium (*Arlt et al., 2018*; *Cates, 2012*; *Frangipane et al., 2018*). As a consequence, the cell density increased significantly ($\varphi \simeq 0.42$ to $\varphi \simeq 0.7$ in 2 min; *Figure 4c*). For the first ~2 min, the average drop in cell speed was likely compensated by the density increase. It is well known, in fact, that cell density can enhance swarming motility through cooperative raft formation (*Be'er and Ariel, 2019*; *Jeckel et al., 2019*). Eventually, however, the increase in cell density resulted first in the formation of jams and finally of multilayer islands (*Figure 4c*, *Video 7*). *Figure 4d* shows the trajectories followed by the irradiated swarms in phase space. Jamming of swarming cells, the first step in island formation, occurs only for cell densities within a range that compares very well with predictions by MIPS for self-propelled rod-like particles (*Figure 4d* shaded region and *van Damme*

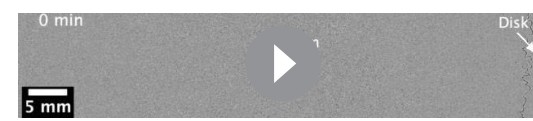

**Video 6.** Formation of multilayers in a swarming plate with a kanamycin-diffusive disk. The multilayer starts forming in the region near the disk where multiple islands appear. The islands become very elongated at the front and they grow in size until they merge. The nucleation and merging of islands were observed up to four layers before we lost the ability to recognize further transitions due to a flattening of the image contrast. In the opposite side of the plate, away from kanamycin, the islands appear and merge homogeneously in the whole region.
https://elifesciences.org/articles/62632#video6

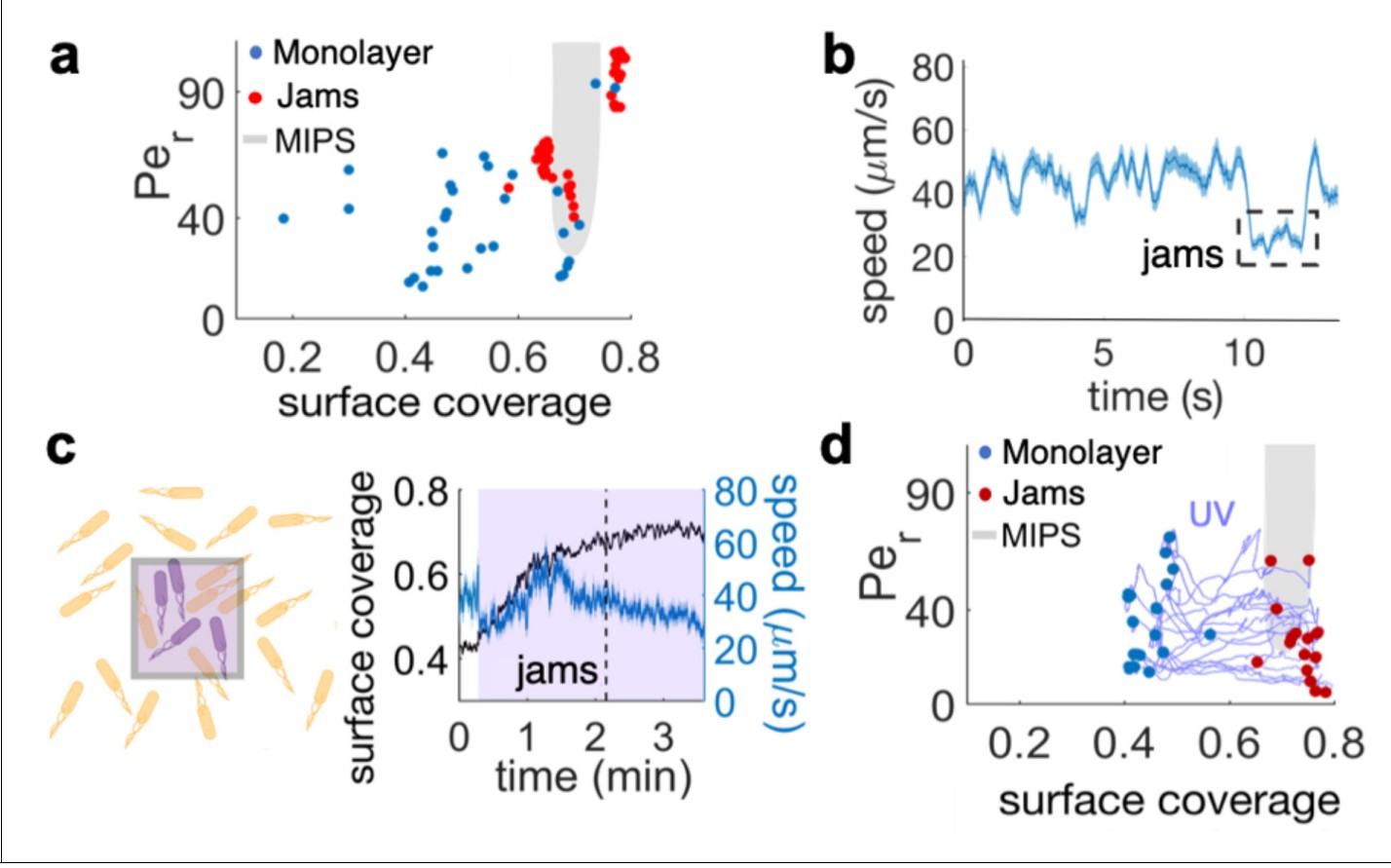

**Figure 4.** The interplay between cell density and cell speed primes jamming and multi-layer formation. (a) Phase diagram of surface coverage and the rotational Péclet number ($P_{e_r}$). $P_{e_r}$ is proportional to cells' speed (see Materials and methods). Grayed region depicts motility-induced phase separation (MIPS) heterogeneity. Each data point represents a result of a 4-s single-cell time-lapse microscopy data. The time-lapse data in which jamming events are observed are shown in red (see *Video 3*). Jams appear exclusively under the conditions near the border of the MIPS region. (b) The speed of motile cells drops by half during jamming (highlighted by dashed rectangle). Increase in jamming events lead to formation of islands (see *Figure 2* and *Video 4*). (c, left) Illustrative diagram showing the accumulation of cells by UV irradiation. Cell speeds drop within the irradiated region (show by purple square), which elevates the surface coverage. (c, right) UV irradiation elevates surface coverage. The graph shows the dynamics of surface coverage (black) and cell speed (cyan). Magenta is the period with UV light illumination (1.2 mW/mm²). Vertical dashed line shows the onset of jamming events. (d) Time evolution trajectories of surface coverage and rotational Péclet number with UV illumination experiment. Blue dots are before UV illumination and red dots are when jams appear. Increasing surface coverage by UV induces formation of jams at the border to MIPS region (gray).

The online version of this article includes the following figure supplement(s) for figure 4:

**Figure supplement 1.** Exposure with UV light of an area much larger than the field of view.

*et al., 2019*). These results provide a direct support to the hypothesis that a biophysical MIPS-like process underpins the transition from swarming monolayers to multilayers in *B. subtilis*.

To further examine if a local cell-density increase is sufficient by itself to induce a localized transition from monolayer to multilayer, we used a physical barrier to impede the advance of the swarm and locally increase cell density. Again, consistently with the MIPS picture, the arrest of the swarm front led to an increase in cell density and the subsequent emergence of multilayer islands near the barrier (*Video 8*). After 36 hr of further incubation, wrinkles developed near the physical barrier precisely in the region where the islands had started to appear initially (*Figure 5a*). As for the kanamycin case, also for the barrier the development of wrinkles is accompanied by a stronger local expression of *tasA* (*Figure 5b* and *Figure 5—figure supplement 1*).

Altogether, these results show that *B. subtilis* swarms can undergo a single-to-multi-layer transition driven principally by a physical mechanism compatible with MIPS, which can be either global and spinodal-like, or localized and binodal-like. When this transition is localized, regardless of it

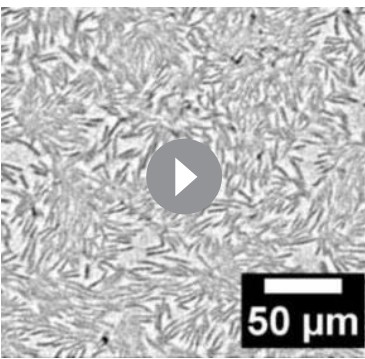

**Video 7.** Time-lapse of accumulation of bacteria over time when they are being continuously irradiated by UV light in a region that is as large as the field of view. Eventually, bacteria accumulation gives rise to the formation of localized islands, which appear as dark regions in the field of view, finally leading to a confluent second layer.

https://elifesciences.org/articles/62632#video7

being caused by antibiotics or physical confinements, the resultant macroscopic cell-density heterogeneity could determine the emergence of wrinkled biofilms.

## Sequential administration of antibiotics reduces the emergence of biofilms from swarms

While more complex signaling pathways regulating biofilm matrix production are likely involved in the passage from localized multi-layers to wrinkles, our results suggest that simply altering the expansion dynamics of a swarm promotes biofilm formation. This implies that exposing bacterial swarms to stressors, such as kanamycin, a physical barrier or UV light, may inadvertently increase their resilience by promoting the formation of biofilms that are much more difficult to eradicate. This is difficult to prevent by simply increasing the amount of antibiotics. In fact, when we used a ~ sevenfold greater dose of kanamycin in the diffusion disk (200 µg), wrinkles still appeared on the plate, although at a greater distance from the disk (*Figure 2—figure supplement 4*). However, our findings suggest that the multi-layer band could be a good target for antibiotic treatment aimed at suppressing the emergence of biofilms. Such multilayer band happens at a concentration of antibiotic that bacteria can tolerate since the cells are still motile (*Figure 3—figure supplement 2*, *Video 5*). We therefore wondered if a two-step sequential antibiotic administration could prevent biofilm formation, where the first administration induces multilayer formation and the second targets the multilayer area before completing biofilm formation. To test this conjecture, we decided to administer a total amount of 200 µg of kanamycin in two steps, an initial one when placing the disk on the plate, and the second as the swarming front stopped. We used a variation of the Kirby-Bauer assay to estimate the kanamycin profile over time at the regions where wrinkles would form and confirmed that the kanamycin concentration at the wrinkle location is compatible across the different dosage protocols (*Figure 6a*, *Figure 6—figure supplement 1*). The emergence of wrinkles was greatly inhibited when kanamycin was administrated sequentially, despite keeping constant the total amount of antibiotic added to the disk (*Figure 6b,c* and *Figure 6—figure supplement 2*). The effect was most evident when the second dose was greater than the first. These results thus propose a promising strategy for treating bacterial collectives with aminoglycosides while minimizing the emergence of biofilms.

## Discussion

This work reveals a biophysical mechanism underpinning the initial stage of the collective stress-induced transition from swarms to biofilm

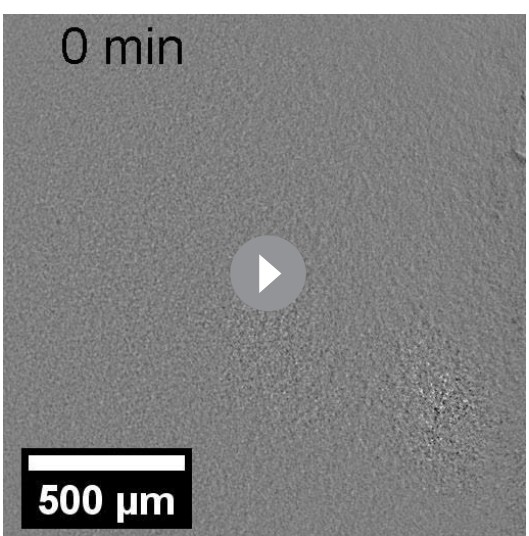

**Video 8.** Process of island formation when the swarming front hits a physical barrier which is placed just on the right of the field of view. Islands start forming in the region closer to the barrier and then propagate towards the opposite direction. After the initial island formation, we can see that the swarm is still dynamic and keep forming multiple layers through island formation and merging.

https://elifesciences.org/articles/62632#video8

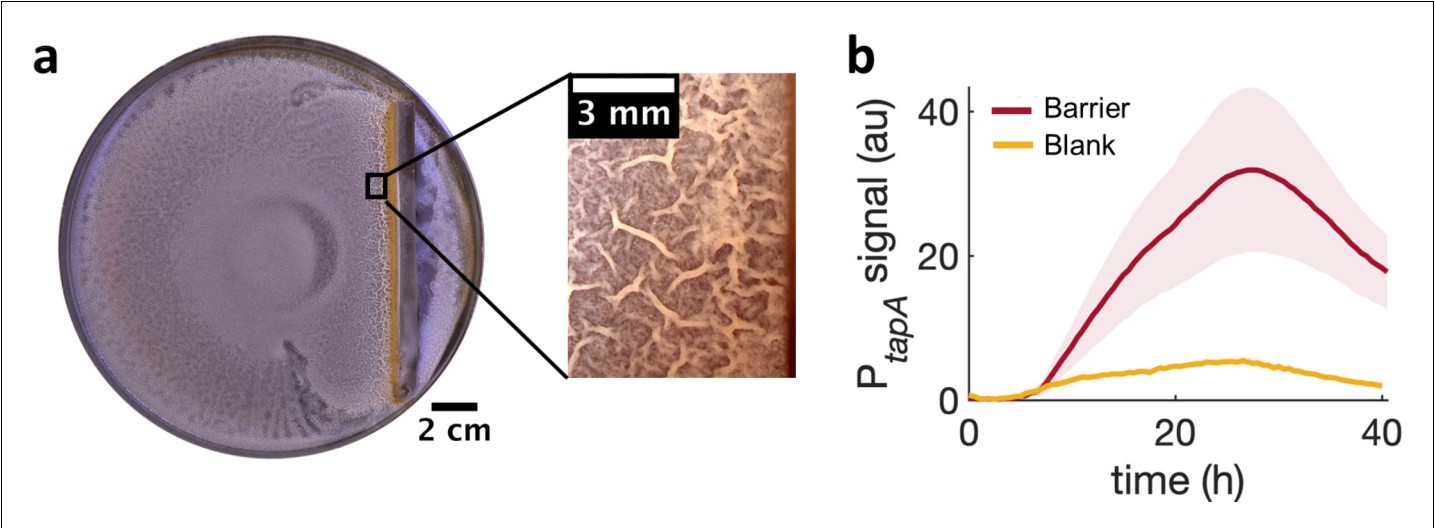

**Figure 5.** Locally increasing cell density is sufficient to induce wrinkle formation. (a) A physical barrier placed on agar triggers wrinkle formation. Wrinkles are formed at the region close to the barrier. A plate was incubated for 40 hr as in *Figure 1*. (b) Mean fluorescence intensity of P$_{tapA}$-*yfp* measured just by the barrier for four different experiments and for two different experiments in the blank case. The shade indicates the s.e.m. The online version of this article includes the following figure supplement(s) for figure 5:

**Figure supplement 1.** P$_{tapA}$-*yfp* expression for four different regions across a swarming plate with a physical barrier in it.

in *B. subtilis*, governed by motility and cell density. This is based on the halting of swarming expansion, which promotes the accumulation of cells at the front, resulting in the formation of MIPS-like multi-layered islands. Upon exposure to the aminoglycoside kanamycin, the swarming colony expresses the *tapA-sipW-tasA* operon and eventually develops a wrinkled biofilm. Consistent with this view, we demonstrate that different stressors, from antibiotics to UV and physical confinement, can all induce formation of islands. Moreover, based on our findings, we show that a sequential antibiotic monotherapy can be effective in reducing biofilm formation from a swarming colony in *B. subtilis*. As the underpinning mechanism of the transition is an emergent phenomenon driven by physical interactions between swarming cells, we believe similar transitions should also happen in other bacterial species. It would be interesting, for example, to examine if swarms of clinically relevant bacteria, such as *Pseudomonas aeruginosa* and *Salmonella enterica*, may also transit into biofilms through a similar process. Such investigations would be an important step forward to see if sequential antibiotic administrations could be effective in inhibiting, and eventually altogether preventing, stress-induced biofilm formation in pathogenic bacteria.

This study addresses a fundamental question about the mechanism by which cell collectives adapt their behavior in response to various physical and chemical stresses. In the present case, a local cell-density increase caused by the halting of the swarming front, may be part of a general collective stress response mechanism, which triggers a switch in the collective behavior from swarming to biofilm. For expansion in presence of kanamycin, these effects might compound with the anomalous motility that has been reported for prolonged exposure to low levels of the antibiotic (*Benisty et al., 2015*). The stress response mechanism that we observe at the collective level could allow the swarming colony to develop biofilms in response to various stressors, regardless of the stressors' exact molecular mode of action. We expect that further research will determine whether this form of environmental sensing and adaptation of cell collectives via cell-density increase is common to other biological systems. Interestingly, this idea is in line with recent advancements in the understanding of mechanochemical feedback in development and disease, where local cell density can determine the fates of cell collectives (*Hannezo and Heisenberg, 2019*). The connection we discovered could represent a primitive example of a collective mechanochemical feedback loop, underpinned by one of the most fundamental types of emergent phenomena (MIPS) in collections of motile agents either alive or synthetic. To this end, the gained biophysical insights may not only offer

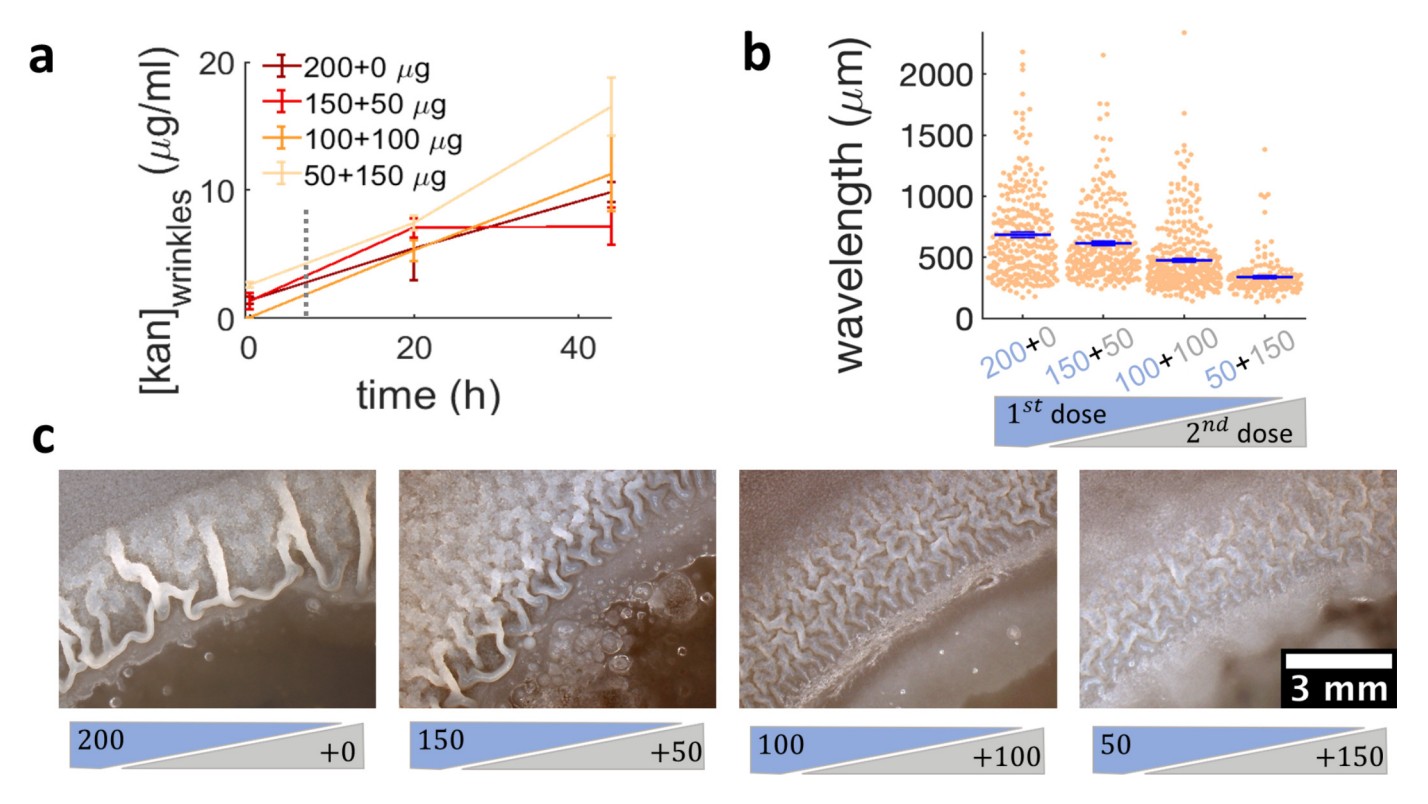

**Figure 6.** Administering the same amount of kanamycin in two sequential doses reduces wrinkle formation. (a), (b), (c) Kanamycin was sequentially administered in two different doses to a disk while keeping the total amount at 200 µg. The first administration was added as in other experiments, while the second was added to the disk as a 4 µl drop when the islands were about to appear. (a) Concentration profile of kanamycin over time for the four concentrations of the sequential administration of antibiotic assay. The measurements were made using the modified Kirby-Bauer assay (see Materials and methods) at the distance from the kanamycin disk where the wrinkles are formed for the four different initial concentrations: 1.32 cm for 50 µg, 1.59 cm for 100 µg, 1.74 cm for 150 µg, and 1.85 cm for 200 µg. The time when the second dose of antibiotic is added is indicated with a dotted gray line (~7 hr after inoculation). (b) The wavelengths of wrinkles with respect to the doses of first (blue font) and second (gray font) administrations (µg). (c) Microscopy images of wrinkles induced by kanamycin. The first dose is in the blue shade and the second dose in the gray shade.

The online version of this article includes the following figure supplement(s) for figure 6:

**Figure supplement 1.** Kanamycin concentration profiles for the four administration protocols (200 µg-0µg; 150–50; 100–100; 50–150) at increasing distances from the kanamycin disk along the radius connecting the inoculation point to the antibiotic disk.

**Figure supplement 2.** Effect of sequential administration of kanamycin in wrinkles' roughness.

new biomedical treatment strategies against the rise of biofilm-associated antimicrobial resistance but may also contribute to our understanding of development and cell-fate determination.

Following our discovery of stress-induced swarm-to-biofilm transition, we present a detailed characterization of the initial stage of the transition, namely development of multilayered islands. As with every discovery, this also brings a host of new questions. For example, it is unclear whether the molecular signaling mechanisms driving biofilm development from planktonic cells are identical to those from swarms. We show the activation of *tasA* gene during the kanamycin-induced transition from swarms to biofilms, suggesting the commonality between these processes. However, our data is inconclusive on this point in part due to the lack of single-cell level analysis of the expression dynamics. The expression of biofilm matrix genes is regulated by various complex pathways, where Spo0A, SinR and AbrB being central regulators (*Vlamakis et al., 2013*). If swarms develop biofilms through different pathways, elucidating the molecular regulatory machinery during the transition from swarm to biofilm may unveil new molecular pathways regulating biofilm formation.

From the perspective of biofilm being a multicellular adaptation, it would be interesting to determine the stage at which a swarming collective loses the ability to adapt to environmental changes. The biophysical transition that we report here would suggest that this happens once the ageing

swarm cannot be driven anymore to within the putative MIPS region by the external stressor. It would be useful to explore this further within the general framework of biophysical models for cluster formation (*Be'er et al., 2020*; *Worlitzer et al., 2020*), 3D architecture formation (*Partridge et al., 2018*) and motility-induced buckling within a bacterial swarm (*Meacock et al., 2020*; *Takatori and Mandadapu, 2003*). Another important question lies in the interplay between biophysical and molecular mechanisms regulating stress-induced biofilm development. Characterizing the gene expression profiles in the high-cell-density clusters resulting from multilayered islands, while simultaneously monitoring the mechanical interactions, would be an important step forward toward gaining a holistic understanding of collective stress response. We hope that our work will inspire new research in this area and look forward to further exciting results in the near future.

# Materials and methods

## Key resources table

| Reagent type (species) or resource | Designation | Source or reference | Identifiers | Additional information |
|---|---|---|---|---|
| Strain, strain background (*Bacillus subtilis*) | NCIB3610 | PMID:20374491 | | Gift from Süel lab |
| Genetic reagent (*Bacillus subtilis*) | *epsH*:: tet$^R$ | PMID:11572999 | | Gift from Kolter lab |
| Genetic reagent (*Bacillus subtilis*) | *phrC*::neo$^R$ | PMID:23012477 | | Gift from Süel lab |
| Genetic reagent (*Bacillus subtilis*) | *oppD*::cm$^R$ | PMID:23012477 | | Gift from Süel lab |
| Genetic reagent (*Bacillus subtilis*) | P*tapA-yfp*, *sacA*::P$_{yqxM}$-*trsA*$_{216}$-*yfp*, cm$^R$ | PMID:26196509 | | Gift from Süel lab |
| Other | Mounted LED | Thorlabs | M405L4 | 405 nm, 1000 mW |
| Other | Aspheric Condenser Lenses | Thorlabs | ACL25416U-B | Ø1', f = 16.0 mm |
| Other | 25 mm SPUTTERED EDGEPASS FILTER | Thorlabs | FELH0450 | Longpass 450 nm |
| Other | Retaining Ring | Thorlabs | CMRR | C-Mount (1.00'−32) |
| Other | Cage cube | Thorlabs | Cm1-dch | Dichroic Filter Mount |
| Other | Adapter | Thorlabs | Sm1a9 | External C-Mount Threads and Internal SM1 Threads |
| Other | Leica DMI Microscope Camera Port Adapter | Thorlabs | Sm1a50 | Internal SM1 Threads, External SM2 Threads |
| Other | Coupler, External Threads | Thorlabs | Sm1t2 | 0.5' Long |
| Other | Adapter | Thorlabs | Sm1a10 | External SM1 Threads and Internal C-Mount Threads |

## Kanamycin gradient assay

Glycerol stock of *Bacillus subtilis* NCIB 3610 wild-type strain (WT) was streaked on a lysogeny-broth (LB) 1.5% agar plate and grown overnight at 37℃. When specified in figure caption, a genetically modified strain (listed in key resource table) was used instead of WT. A single colony was picked from this plate and incubated in 1 ml of liquid LB for 3 hr at 37℃. A 4 µl inoculum from this culture was placed in the center of a 0.5% LB agar plate supplemented with 2% of glycerol and 0.1 mM MnSO4 (LBGM *Shemesh and Chai, 2013*) to favor biofilm formation. A kanamycin-diffusive disk (Oxoid 30 µg) was placed on a side of the plate 24 hr before inoculation to allow the antibiotic to diffuse at room temperature. The distance between the inoculum and the kanamycin disk was approximately 3.2 cm, and plates were incubated for additional 40 hr after inoculation at 30℃.

## Images of swarming plates

Low magnification images of the plates were acquired with a DSLR D5000 Nikon camera (lens AF-S Micro NIKKOR 40 MM 1.28) with an inhouse stand in a 30°C incubator (*Kantsler et al., 2020*). The incubator was covered by black tape to avoid reflections and the illumination was provided by a white LED placed on a side of the plate. Higher magnification images (e.g. of the wrinkles) were taken by an Olympus SZ61 microscope by placing the plates in a dark background with illumination coming from a LED ring attached to the microscope.

## Quantification of the $P_{tapA}$-*yfp* reporter

Biofilm extracellular matrix production was characterized by using a modified strain carrying $P_{tapA}$-*yfp*, a fluorescence reporter for the expression of *tapA-sipW-tasA* operon. The kanamycin gradient assay and the barrier experiments were repeated by inoculating this strain in LBGM (1.2% and 2% glycerol) swarming agar plates. The experiment was replicated three times for the kanamycin case with three different lenses and microscopes: 2x (Nikon Apo Lambda 2x UW, NA 0.1), 2x (Nikon Plan 2x UW, NA 0.06), 2.5x (Leica 2.5x N PLAN, NA 0.07); microscopes: two Nikon Eclipse Ti2 and a Leica DMi8. For the barrier case the experiment was repeated four times and it was performed in the Nikon Eclipse Ti2 using 2x (Nikon Plan 2x UW, NA 0.06). Images were taken every half an hour for a period of ~40 hr in a region of $3 \times 9$ cm$^2$ going from the disk to the inoculum. The images were stitched using the 'Grid/Collection stitching' plugin (*Preibisch et al., 2009*) in Fiji (*Schindelin et al., 2012*). The experiment was repeated three times in absence of kanamycin with the first two microscopes and lenses. To calculate the fluorescent signal, a region of interest was drawn in Fiji surrounding the area where the wrinkles appeared, normally found just at the edge of the depletion region created by the kanamycin disk or just right next to the physical barrier. For the kanamycin assay, the signal was normalized by subtracting the minimum value of pixel intensity recorded in the time-lapse and dividing by the maximum of the signal for each of the microscopes to account for the difference in signal of each microscope. In the barrier case, all the experiments were performed with the same microscope and lens, so the raw signal is shown.

## Raft and islands sizes

The size of the rafts within the swarm were obtained from three different experiments. A freehand line drawn with Fiji/imageJ (*Schindelin et al., 2012*) enclosing the raft was used to measure the area within the line. Measurements across different positions and time points were used to account for the variability in raft size within a swarm depending on the position and/or time after expansion begins (*Jeckel et al., 2019*). To measure the islands size, time-lapses of islands formation in presence and absence of kanamycin were used. The first frame (just before islands appeared) was subtracted to the time-lapse and then a gaussian filter was applied to remove noise. Finally, the time-lapse was thresholded and the initial size of the islands was measured by 'regionprops' in Matlab.

## Characterization of island formation

We characterized the double layer using 2x (Nikon Plan 2x UW, NA 0.06), 2.5x (Leica 2.5x N PLAN, NA 0.07) and 10x (Nikon 10x PLAN FLUOR PH2 DLL, NA 0.3) magnifications in two different microscopes, Nikon Eclipse Ti2 and Leica DMi8. The images were acquired every 2 min in the Nikon Eclipse Ti2 and every 1 min in the Leica Dmi8 using brightfield illumination. Cell motility and cell density were recorded with a 40x objective (Nikon 40x LWD, NA 0.55) at 37°C. Cell motility was measured by adapting a Particle Image Velocimetry (PIV) code written in Matlab (*Sveen, 2004*). The surface coverage was measured by thresholding the images and dividing the area covered by bacteria by the total area of our field of view. The threshold was set by using the command 'imbinarize' in Matlab and adapting the sensitivity of its threshold to account for the best estimate of cells in the field of view.

## Near UV experiments

Bacteria were irradiated by near UV-Violet light for 30 s using the inverted microscope DMi8 (Leica Microsystems) and the LED light source, SOLA SM II Light Engine (Lumencor), with an excitation filter 400/16 using a 40x objective (Leica 40x PH2 HC PL FLUO, NA 0.6) for approximately 3.5 min at an intensity of 1.2 to 5.3 mW/mm$^2$. The light intensities were measured by placing a photodiode

power sensor (Thorlabs S120C) on the microscope stage. To irradiate a larger area than the one with the epi-fluorescence set up, a Thorlabs 405 nm light LED was coupled to a cage cube (see key resources table for details about the components of the set up). The light coming from the LED was concentrated by an aspheric condenser lens and reflected by a 45˚ dichroic mirror towards the sample. The area illuminated is roughly a circle of 2.5 mm radius at an intensity of 1.5 mW/mm$^2$. The surface coverage was calculated by binarizing the time lapses applying a locally adaptive threshold in Matlab. The sensitivity of such threshold was changed along the time lapse when cell density was rapidly increasing and the same value of the threshold could not account for the total number of particles.

The trajectories of the swarming bacteria ensemble in the phase diagram were extracted by calculating the Pe$_r$ and the surface coverage as detailed in the previous section. In the case of UV exposure, trajectories are the result of continuous irradiation with either of the setups previously described. When illuminated by the epifluorescence the trajectories were plotted until a jamming of bacterial rafts was observed in the field of view. When shining UV from the condenser the trajectories were plotted until bacteria were completely immotile, as jamming events never happened (*Figure 4—figure supplement 1*). Data for speed and surface coverage of bacteria has been smoothed over 8 and 100 points respectively by the 'smooth' function in Matlab.

## Quorum-sensing experiments

The same protocol explained in the section on the 'kanamycin gradient assay' was also used to investigate the swarming behavior, the formation of islands and the biofilm development of the two quorum-sensing knockout strains: Δ*opp* and Δ*hprc*. Formation of islands and biofilm images were acquired as explained in previous sections.

## Phase diagram

To calculate the phase diagram of rotational Péclet number (Pe$_r$) with respect to surface coverage, time-lapses of swarming bacteria coming from six different experiments were analyzed. The 'jammed bacteria' data plotted in the diagram (*Figure 4a*) were calculated by analyzing the 4 s of the time lapse prior to jam formation. Videos were acquired at 29–33 fps.

The average speed within the swarm was calculated as described earlier and used to then calculate the Pe$_r$, defined as:

$$Pe_r = \frac{u}{LD_r}$$

where $u$ is the average cell speed within the swarm for a given time point, $L$=0.823±0.16 μm the average width of the swarming cells -as used in *van Damme et al., 2019*- and $D_r$ is the rotational diffusivity of the bacteria. For swarming cells, the rotational diffusivity cannot be measured directly. Therefore we decided to use, as estimate for $D_r$, a value such that the Péclet number of our system and that in *van Damme et al., 2019* coincide at surface coverage 0.65. This returns $D_r$ = 0.81 ± 0.19 s$^{-1}$, from an average of 13 different values from three independent experiments. To check whether this result was sensible, we then resuspended in water a sample of swarmer cells from a plate and extracted the rotational diffusivity of motile cells from a 10 s movie, with a Python-based tracking code which measures position and orientation of elongated objects (*Mosby et al., 2020*). The gradient of the angular mean-square displacement of the tracks returned a value of $D_r$ = 0.35 s$^{-1}$. Although this is slightly lower than the previous result, it should be kept in mind that the former estimate is for cells that are moving on an agar surface, which experience a higher drag than those swimming in a bulk fluid.

Notice that, Pe$_r$ in *van Damme et al., 2019* is obtained from instantaneous velocities of individual particles, which cannot be directly followed within a swarm. Therefore, we used an effective Pe$_r$ derived from the local average velocity within the swarm. This is likely to overestimate the value of Pe$_r$ of the experimental points.

## Physical barrier

A 3% agarose solution in water was autoclaved and then poured in a Petri dish. Once it solidified, a rectangular region (6 cm x 1 cm) was cut out and placed vertically on a molten swarming liquid

LBGM (0.5% agar). After the system solidified, swarming bacteria were inoculated in the centre of the plate. Videos of the formation of islands were recorded under 2x (Nikon Plan 2x UW, NA 0.06) in a Nikon Eclipse Ti2 microscope.

### Biofilm inhibition assay

A total of 4 µl of the antibiotic kanamycin coming from four different stocks at concentrations of 50, 37.5, 25, and 12.5 µg/µl were added to four different diffusive disks. The diffusive disks were then placed on the side of different LBGM 0.5% agar plate for 24 hr. After bacteria inoculation, the plates were incubated at 30°C for roughly 6 hr until the swarming front halted. Then, additional 4 µl of the second stock concentrations of kanamycin were added to the initial ones. For the four concentrations above, these were respectively 0, 12.5, 25, and 37.5 µg/µl. As a result, the total amount of kanamycin administered was kept constant and equal to 200 µg.

### Estimation of the kanamycin profile on the swarming plate

The concentration profile of the kanamycin in the plate (*Figure 5a*) was characterized by using a variation of the Kirby-Bauer assay (*Bauer et al., 1966*). A total of 800 µl of a 18 hr bacteria culture at 30°C was inoculated in three plates containing LB with 1.5% agar. Immediately after, disks containing 0.1, 0.2, 0.4, 0.78, 1.56, 3.13, 6.3, 12.5, 25, 50, 100, 150, and 200 µg of kanamycin were placed on the surface of separate confluent LB agar plates. After an incubation of 24 hr at 37°C, we measured the diameter of the clearance region around the disks. Each measurement was done in triplicates. The logarithm of the mass of kanamycin added was plotted against the difference between the diameter of the clearance region and the diameter of the disk. This calibration curve was then used to characterize the kanamycin profile in both the swarming and the biofilm-inhibition assays. For each replica agar plate, a 3 mm diameter biopsy puncher was used to extract 5 small samples in a range of 0–2 cm every 0.5 cm distance from the kanamycin diffusive disk. The agar punches were then placed on top of a confluent plate of WT *B. subtilis* and incubated overnight at 37°C. We then measured the clearance around the pads and used the linear fit to the calibration curve (*Bonev et al., 2008*) to estimate the (logarithm of) total amount of kanamycin within the punch. This was then divided by the volume of the punch to estimate the concentration.

### Wrinkle wavelength quantification

To measure the wavelength of the wrinkles we calculated the interdistance of nearest parallel wrinkles. When this was not possible, the wavelength was estimated by calculating the autocorrelation function of image intensity in space using a Fiji macro made by Michael Schmid (https://imagej.nih.gov/ij/macros/RadiallyAveragedAutocorrelation.txt) (2008) and then fitting the decay of that function to a double exponential of the form:

$$f(x) = ae^{bx} + ce^{dx}.$$

Here, the two characteristic wavelengths (1/b and 1/d) correspond to the image noise and the actual wrinkle wavelength respectively.

### Calculation of the minimum inhibitory concentration (MIC)

A single colony of *Bacillus subtilis* was incubated in 1 ml LB overnight in a shaking incubator at 37°C. The OD was measured 18 hr later and the culture was diluted down to an OD of 0.0085 and incubated for 18 hr in 5 concentrations of kanamycin from 1 to 5 µg/ml. The MIC was determined as the minimum concentration at which growth was not observed in the liquid culture. Each well plate had five replicates and the experiment was repeated at least three times for each concentration.

### Roughness measurements for wrinkles

A 50 pixels wide strip was drawn over the wrinkles and then the intensity profile was extracted using 'Plot profile' in ImageJ. The roughness of the profile was defined as:

$$roughness = \frac{\sum_{i}^{N} \left| I_i - \bar{I} \right|}{N},$$

where $I_i$ is the intensity of $i$th pixel in the profile, $\bar{I}$ is the mean of the profile obtained using the 'smooth' function in Matlab and $N$, the total number of points in the profile.

## Quantification of the speed at the swarming front

To calculate the speed at the swarming edge (*Figure 2—figure supplement 2*), the position of the front closest to the diffusive disk was recorded over time with a DSLR camera (Nikon D5300; lens AF-S Micro NIKKOR 40 MM 1.28) at 37°C. Consecutive front positions were subtracted to highlight the position of the front. This was then used to estimate the front's speed on both the kanamycin side and the opposite side of the agar pad using the inter-frame time of 5 min. From these time lapses, we also extracted the distance at which the swarming front halts from the kanamycin disk.

## Multilayer formation within the swarm

To investigate the spatial difference in multilayer formation throughout the swarm, videos of the bacteria were taken at ×2 magnification and one frame per minute, for 2 hr at 30°C (*Video 6*). The videos were acquired following a tiling of the rectangular region from the diffusive disk to the opposite end of the plate, with 12 sectors along the diameter and five across. The videos were then stitched together using the 'Grid/Collection stitching' plugin (*Preibisch et al., 2009*) in Fiji. To enhance the contrast of the different layers, the frame just before the multilayer appeared was subtracted from the rest of the time-lapse. The different layers appearing in the swarm were counted for four different positions and they were identified by a local increase in gray value in a wide area of the swarm (*Figure 3—figure supplement 3*). After the fourth layer, the increase in gray value could not be further identified. Time is measured from the onset of the first islands within the field of view.

## Distance from wrinkles to kanamycin disks of different concentration

To characterize how the distance between the kanamycin disk and the wrinkles depends on the initial antibiotic concentration within the disk (*Figure 2—figure supplement 4*), diffusive disks containing 30, 100, and 200 µg of kanamycin were placed on 2-3% glycerol LBGM agar plates. To create the disks, Whatman filter paper was punched with a hole puncher (diameter 7 mm) and imbibed with 0.6, 2, or 4 µl of a 50 µg/µl kanamycin solution in water. The minimum distance between the kanamycin disk and the nearest wrinkles was measured in six different experiments for 30 µg and 200 µg of antibiotic and one experiment with three replicates for 100 µg.

## Biofilm formation in solid agar near the kanamycin

To check whether wrinkle formation was promoted also in absence of motility near the kanamycin diffusive disk, the kanamycin-diffusive assay was repeated in 1.5% agar. To do so, a kanamycin-diffusive disk was place on one side of the plate of the solid agar LBGM (2% glycerol) plate. The position was chosen to match the disk position in the swarming experiments. After 24 hr, a single colony of *B. subtilis* was incubated in 5 ml of LB for 3 hr. Of this culture, 700 µl was spread on each solid agar plate and then incubated for 40 hr at 30°C.

We also check the wrinkle formation in 1.5% agar by spotting 4 µl of the bacteria culture without spreading them to see if wrinkles were promoted in the wrinkly biofilms that appear when bacteria are spotted in hard agar. In total, four different colonies were spotted on the plate: the closest one was spotted 2 cm away from the kanamycin disk, other two spots were placed 2 cm to the right and the left of this central spot and finally, a last colony was spotted 6 cm away from the kanamycin disk (*Figure 2—figure supplement 6*).

## Measuring the width of the band where wrinkles appear

The width of the band where wrinkles appeared around the kanamycin disk was measured for three independent experiments using 30 and 200 µg of kanamycin. The width was determined by drawing a line across the band where the wrinkles appeared.

## Statistics

Data are reported as Mean± s.e.m. calculated from at least three independent experiments unless otherwise indicated.

## Acknowledgements

This research is funded by the MRC Doctoral Training Partnership (MR/N014294/1). MP and MA acknowledge support from EPSRC grant, Bridging the Gaps initiative (EP/M027503/1). MA acknowledges BBSRC/EPSRC grant to the Warwick Integrative Synthetic Biology Centre (BB/M017982/1). MP acknowledges support from a Ramón y Cajal Fellowship (RYC-2018-025345-I). We thank Matthew Painter and Chris Norman for their help in the first experimental stages of the project, Lewis Mosby for his help with estimates of the rotational diffusivity, the Nelson lab for use of their stereomicroscope, the Süel lab and the Kolter lab for the strains, and Drs. Darius Köster and Meera Unnkrishnan for their comments to the manuscript.

## Additional information

### Funding

| Funder | Grant reference number | Author |
| --- | --- | --- |
| Medical Research Council | MR/N014294/1 | Iago Grobas |
| Engineering and Physical Sciences Research Council | EP/M027503/1 | Marco Polin Munehiro Asally |
| Biotechnology and Biological Sciences Research Council | BB/M017982/1 | Munehiro Asally |
| Ministerio de Ciencia e Innovación | RYC-2018-025345-I | Marco Polin |

The funders had no role in study design, data collection and interpretation, or the decision to submit the work for publication.

### Author contributions

Iago Grobas, Conceptualization, Formal analysis, Investigation, Visualization, Writing - original draft, Writing - review and editing; Marco Polin, Munehiro Asally, Conceptualization, Supervision, Funding acquisition, Investigation, Visualization, Writing - original draft, Writing - review and editing

### Author ORCIDs

Iago Grobas  https://orcid.org/0000-0003-3609-8081
Marco Polin  https://orcid.org/0000-0002-0623-3046
Munehiro Asally  https://orcid.org/0000-0002-8273-7617

### Decision letter and Author response

Decision letter https://doi.org/10.7554/eLife.62632.sa1
Author response https://doi.org/10.7554/eLife.62632.sa2

## Additional files

### Supplementary files

• Transparent reporting form

### Data availability

All data generated or analysed during this study are included in the manuscript and supporting files.

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
