## [Decision Letter]

**Acceptance summary:**

The paper presents very interesting results linking bacterial swarming to biofilm formation. It shows that swarming bacteria can initiate biofilm formation through a motility induced phase separation process.

**Decision letter after peer review:**

Thank you for submitting your article "Swarming bacteria undergo localized dynamic phase transition to form stress-induced biofilms" for consideration by *eLife*. Your article has been reviewed by two peer reviewers, and the evaluation has been overseen by a Reviewing Editor and Aleksandra Walczak as the Senior Editor. The reviewers have opted to remain anonymous.

The reviewers have discussed the reviews with one another and the Reviewing Editor has drafted this decision to help you prepare a revised submission.

Summary:

This manuscript is based around a potentially very interesting finding that links bacterial swarming to biofilm formation. The authors suggest that swarming bacteria can initiate biofilm formation through a mobility induced phase separation (MIPS) like process. The authors demonstrate that biofilms form near areas of the plate where a kanamycin gradient is placed. They put forward the hypothesis that this is caused by a MIPS transition.

Both reviewers found the paper interesting, and results potentially relevant. However, they have raised a few major concerns.

Essential revisions:

i) Even though the data, in terms of jamming and island formation, appears consistent with a MIPS transition, there is less evidence that this transition plays a role in the formation of biofilms.

ii) Some of the claims based on the experimental results are not fully justified and additional control experiments are required.

The reviewers have listed a number of important issues to be addressed and have suggested experiments/analysis to solve them. All these issues must be solved before publication. Please do all the changes/additional work the reviewers ask for and reply to all their comments.

Reviewer #1:

The authors do not actually show that kanamycin reduces cell motility, even though this is a core assumption of their interpretation. Can the authors directly measure that the addition of certain concentrations of kanamycin reduce *B. subtilis* flagella-based motility (e.g. in liquid culture)? This appears to be a key control experiment. Furthermore, how do the authors rationalize that an aminoglycoside could reduce swarming motility?

Given that the swarm cannot expand beyond the swarming plate, the walls of the petri dish should also pose a barrier to it. Could the authors comment on why no biofilm formation can be observed in the case of a swarm growing in a regular petri dish, as they show in Figure 2D, E? Does this mean that a particular geometry of the barrier is required in order to trigger biofilm formation?

For the experiments using a physical barrier as well as a two-step antibiotic treatment, the authors use the existence of wrinkles and their wavelength as a measure for biofilm formation. These measures are more a quantification of the biofilm structure/architecture/morphology, and not a quantification of the biofilm formation capability. The more appropriate metric for quantifying biofilm amount is the biofilm biovolume or biomass (i.e. non motile cells that are bound in an extracellular matrix). Could the authors please modify their language accordingly, or provide a measurement of the biofilm biomass in some form?

Regarding Figure 5, and the sequential application of kanamycin. The wording of the Results text is currently not justified by the data. The authors write "sequential administration of antibiotics suppresses the emergence of biofilms from swarms" and "…effective in preventing biofilm formation…". However, the data in Figure 5 only shows that the wrinkling wavelength is reduced. Therefore, the data show that biofilms are not suppressed, they are still there, they just have a different shape. Results in Figure 5 are interesting, but I think that the authors have to walk back their conclusions a little bit here, to more closely align with their data.

The following point may require a better explanation in the Discussion or in the Results: It is unclear to me while sometimes the cells can move on top of each other across the cell layers (see Results), and how the authors on the other hand argue that there is jamming when the cell density gets too high. What cell density then causes jamming if it is not achieved by complete surface coverage? Is it perhaps that once the cells start to produce extracellular matrix, the matrix produces a barrier that causes jamming? More careful wording of the relevant sections is required.

Reviewer #2:

It has been demonstrated for multiple bacteria that sub-inhibitory amounts of antibiotics, including kanamycin, can induce biofilm formation, under growth conditions that do not necessarily support a MIPS hypothesis (e.g. liquid culture (PMID: 24260549)). These previous works are not discussed in the paper and could point to a different mechanism of biofilm formation through antibiotic addition than a MIPS transition through swarming.

The data presented could be explained by a different hypothesis rather than a MIPS transition. For example, just the fact that biofilm formation requires high cell density. The experiments are carried out on biofilm inducing medium – this suggests that only high cell density is needed for biofilm formation and if the authors waited long enough biofilms would form under all the conditions tested (irrespective of antibiotic application or swarming). There is not a great deal of evidence that the MIPS transition is actually playing a role. For example, the authors use growth on hard agar (which does not support swarming) as evidence that swarming is required for biofilm formation. But the hard agar plates are made with biofilm promoting media, so presumably the only reason no biofilms are seen is that the cell density is not high enough. In the hard agar control, at the beginning of the experiment cells are spread across the whole plate.

What happens with an inoculum started in the centre of the hard agar plate (similar to the swarming experiments) and a KAN disc placed nearby. Too many differences exist between the hard agar and swarming experiments (e.g. number of cells, placement of antibiotic disk in different location on dish, method for inoculum placement) to be able to say clearly that swarming is required for biofilm formation near the KAN gradient.

The double antibiotic experiment appears to have an issue. As I understand it the design of the experiment is as follows “We administer a total amount of 200 µg of kanamycin in two steps, an initial one when placing the disk on the plate, and the second as the swarming front stopped (Figure 5B, Figure 6—figure supplement 2). The emergence of wrinkles was greatly suppressed when kanamycin was administrated sequentially, despite keeping the total amount of antibiotic constant (Figure 5B, C and Figure 6—figure supplement 2).”

I am concerned that the authors do not know exactly how much antibiotic the cells are experiencing as the antibiotics will diffuse over time into the plate (as the authors show in Figure 2—figure supplement 1). Are the authors sure that the cells at the edge of the disc are really experiencing the same amount of antibiotic if the antibiotic is added 24 h later? I would assume that the concentrations would be higher for the cells at the swarming front on the second application due to less time for the kanamycin to diffuse (as suggested by Figure 2—figure supplement 1). Could this explain why the second application is so effective? Apologies if this is controlled for and I missed it.

---

## [Author Response]

Reviewer #1:The authors do not actually show that kanamycin reduces cell motility, even though this is a core assumption of their interpretation. Can the authors directly measure that the addition of certain concentrations of kanamycin reduce B. subtilis flagella-based motility (e.g. in liquid culture)? This appears to be a key control experiment. Furthermore, how do the authors rationalize that an aminoglycoside could reduce swarming motility?

The reduction of cell motility in a swarming *B. subtilis* colony by kanamycin was investigated in a seminal paper by Benisty et al. (Benisty et al., 2015). Figure 2B in Benisty et al. shows a linear decrease of the average cell motility in swarming *B. subtilis* with [Kan], ~1 µg/ml. We thank the reviewer for this comment, which made us realise that we had inadvertently left out this reference. It has now been added to the manuscript.

To test whether this is happening also in our kanamycin experiments, which exposes the cells to an antibiotic gradient, we have now measured the average swarming speed at the distance of 0.7, 1.7 and 2.7 cm from the kanamycin disk. Figure 3—figure supplement 1, which follows the speed at these locations from 60 min before islands emerge (-60 min) until their emergence (0 min), shows that a significant reduction in speed happens at all distances, although the trend is less pronounced at the largest separation (2.7cm) probably due to the minute concentration of kanamycin at that position. Notice that, as the front propagates along the kanamycin gradient, from ~1.7 cm at -60 min to ~0.7 cm at -15 min, the speed of cells at the front of the swarm drops from >40 µm/s to 0 µm/s. Within the final 15 min, this happens without an appreciable advance of the swarming front (which is approximately at 0.7 cm). Altogether, the study by Benisty et al. and our measurements of cell speed at the swarming front demonstrate that the cells slow down under kanamycin exposure and can eventually become immotile.

At the same time, we respectfully disagree with the reviewer’s statement that reduction of cell motility by kanamycin is a core assumption of our interpretation. The fact that kanamycin blocks the expansion of the swarming front is important for the emergence of islands because it leads to an increase in cell density within the swarm. However, the reduction in cell motility does not have a direct role in the formation of islands per se. It is important here to distinguish between -on one hand- a general reduction in cell motility like that caused by prolonged exposure to kanamycin; and -on the other hand- the fact that there is a reduction in speed when cells moving within a monolayer take part in a jamming event or become part of a multilayer island. In the second case, the speed reduction is essentially a result of crowding. If/when cells manage to go back to moving within the monolayer, they will go back to their original (high) speed (see e.g. Video 3). MIPS is based on this crowding-induced slowdown (and on the fact that it is self-reinforcing), and *not* on a general speed reduction like the one induced by kanamycin. In fact, our barrier experiments show directly that an exogenously-induced general slowdown of the average speed is not necessary for islands to emerge. Instead, what is important is that movement within the multilayer islands is effectively slower than in the monolayer. This makes sense given the disordered nature of these multi-layered regions, with cells clearly moving across the different layers (see Video 4) and it is also borne out of PIV measurements from high-magnification videos. PIV-based estimates of local cell movement suggest a reduction of about 50% in the 2D speed of cells within an island compared to the surrounding monolayer (see Figure 3—figure supplement 2).

Given that the swarm cannot expand beyond the swarming plate, the walls of the petri dish should also pose a barrier to it. Could the authors comment on why no biofilm formation can be observed in the case of a swarm growing in a regular petri dish, as they show in Figure 2D, E? Does this mean that a particular geometry of the barrier is required in order to trigger biofilm formation?

In fact, we observed that the wrinkles can be formed at the petri boundary both in the experiments with a kanamycin disk and in experiments without kanamycin. An example can be seen in Author response image 1. However, these wrinkles are much fainter than our barrier experiment. While there could be many explanations to this, including barrier geometry, we suspect that this may be due to the agar meniscus on petri boundary. This meniscus is both a complicating factor for swarm visualisation and causes potentially different local conditions in nutrient availability, as the local thickness of the agar is different from the rest of the petri dish. Using a straight barrier was a better controlled experiment for test our hypothesis and show that a physical barrier can indeed promote prominent wrinkle formation as presented in our main figure.

**Author response image 1. respfig1:** Radial transects of a 9 cm-diameter petri dish with a swarming *B. subtilis* culture in conditions corresponding to “no kanamycin” and “kanamycin”, as specified in the manuscript.

For the experiments using a physical barrier as well as a two-step antibiotic treatment, the authors use the existence of wrinkles and their wavelength as a measure for biofilm formation. These measures are more a quantification of the biofilm structure/architecture/morphology, and not a quantification of the biofilm formation capability. The more appropriate metric for quantifying biofilm amount is the biofilm biovolume or biomass (i.e. non motile cells that are bound in an extracellular matrix). Could the authors please modify their language accordingly, or provide a measurement of the biofilm biomass in some form?

While we agree with the reviewer that wavelengths of wrinkles are a morphological feature, their connection with the mechanical properties of biofilms in *B. subtilis* is well established (Asally et al., 2012; Kesel et al., 2016; Yan et al., 2019). Biofilms are defined as a structured community of cells enclosed in extracellular matrix. There is a general consensus that extracellular matrix is an integral part of biofilms. In the case of *B. subtilis,* it has been shown that the colonies formed by the mutants lacking the genes encoding extracellular matrix still develop biomass/biovolume, but do not develop the wrinkled morphology (Branda et al., 2001). As such, we believe that the wrinkled morphology is an immediate and convenient reporter of biofilm development. In the original submission, we commented explicitly on this point in the text: “To quantify the degree of biofilm formation, we measured the characteristic wavelength and roughness of the wrinkles, which have been reported to correlate with biofilm stiffness […]”.

In order to try to be as clear as possible, we also amended the text right after the lines quoted above, which now reads:

“In this study, the term biofilm is interpreted as wrinkly biofilm, and the wrinkles’ wavelength is used as a convenient indirect quantification of biofilm formation since the wavelength has been reported to correlate with biofilm stiffness and extracellular matrix (Asally et al., 2012; Kesel et al., 2016; Yan et al., 2019). More specifically, the wavelength is smaller in colonies of matrix mutants and greater in hyper biofilm-forming mutants.”

Moreover, we would like to draw attention to our experiments with a *tapA* promoter reporter strain to measure upregulation of TasA extracellular matrix component. As seen in Figure 2E, the conditions that lead to the development of wrinkles from the swarm also lead to a strong fluorescence signal, suggesting a higher local production of TasA.

Altogether, we believe that our focus on the wrinkle phenomenology to characterise locally the biofilm is appropriate and justified.

Regarding Figure 5, and the sequential application of kanamycin. The wording of the Results text is currently not justified by the data. The authors write "sequential administration of antibiotics suppresses the emergence of biofilms from swarms" and "…effective in preventing biofilm formation…". However, the data in Figure 5 only shows that the wrinkling wavelength is reduced. Therefore, the data show that biofilms are not suppressed, they are still there, they just have a different shape. Results in Figure 5 are interesting, but I think that the authors have to walk back their conclusions a little bit here, to more closely align with their data.

As noted above, in the case of *B. subtilis* air-interfacing biofilms, wrinkle wavelengths have been shown to correlate with the biofilms’ mechanical stiffness and extracellular matrix abundance (Asally et al., 2012; Trejo et al., 2013).

Nevertheless, we agree with the reviewer that the sequential administration did not completely eliminate the development of wrinkles. We have now amended the text of the manuscript in both places to read:

“Sequential administration of antibiotics reduces the emergence of biofilms from swarms” and “[…] Moreover, based on our findings, we show that a sequential antibiotic monotherapy can be effective in reducing biofilm formation from a swarming colony in *B. subtilis*.[…]”.

The following point may require a better explanation in the Discussion or in the Results: It is unclear to me while sometimes the cells can move on top of each other across the cell layers (see Results), and how the authors on the other hand argue that there is jamming when the cell density gets too high. What cell density then causes jamming if it is not achieved by complete surface coverage? Is it perhaps that once the cells start to produce extracellular matrix, the matrix produces a barrier that causes jamming? More careful wording of the relevant sections is required.

We apologise if the paper was not clear on this point. The jamming we refer to has nothing to do with secretion of extracellular matrix, which happens only much later in the system. Instead, it happens when sufficiently large rafts within the swarming monolayer collide. The collision creates a transient localised region of high cell concentration, characterised by a strong reduction in the average speed of the cells and the protrusion of several cells outward from the monolayer. However, if the overall cell concentration is low enough, these temporary jams eventually disappear, as the cells manage to move away.

We have now included a video showing this phenomenon at 40x magnification (Video 5). We apologise for not including it earlier, despite it being cited in the manuscript. This was due to a trivial mistake in the uploading of the videos in the first submission, which has now been corrected. As seen in Video 4 (revised version), these localised temporary jamming events can also be observed at low magnification as darker regions that appear and then disappear. Notice that this is true both before and after the emergence of stable islands (left and right panels in the video) although in the latter case temporary jams have the tendency to last longer. This is coherent with our understanding of jamming events as are nucleation attempts for islands, similarly to what happens in a standard first order phase transition.

In order to clarify the issue raised by the reviewer, besides including Video 3 we have modified the text to read:

“[…] at which point the swarming rafts started displaying jamming events lasting ~1-2 sec, during which cell speed was strongly reduced and groups of cells protruded temporarily from the swarming monolayer […]”.

Reviewer #2:It has been demonstrated for multiple bacteria that sub-inhibitory amounts of antibiotics, including kanamycin, can induce biofilm formation, under growth conditions that do not necessarily support a MIPS hypothesis (e.g. liquid culture (PMID: 24260549)). These previous works are not discussed in the paper and could point to a different mechanism of biofilm formation through antibiotic addition than a MIPS transition through swarming.

Indeed, it is known that aminoglycosides can induce biofilm formation in *P. aeruginosa* and *E. coli* (Hoffman et al., 2005; Jones et al., 2013). We agree that this is an important point to flag in the manuscript and have now added the paper suggested by the reviewer. While these prior studies investigated biofilm formation from swimming cells, our study was focused on the transition from swarming cells. It is also worth noting that, to the best of our knowledge, the induction of biofilm formation by aminoglycosides has not been demonstrated in *B. subtilis.*

What we observe in the experiments, however, is that kanamycin exposure is not enough to promote biofilm formation per se. We address this point in more detail in our answer to the next question by this reviewer. Instead, at the same time, even a simple barrier can induce the formation of biofilm. This appears through a local cell accumulation which is fundamentally identical to the kanamycin case, as in both cases it results from halting the expansion of the swarming front. This local cell accumulation happens by itself as a result of cell motility through a single-to-multi layer transition that has the hallmarks of MIPS.

Therefore, although we agree with the reviewer that there are multiple avenues to the formation of biofilms, we believe that our study reveals a general novel mechanism that is not just the result of exposure to antibiotic, but happens instead as a result of collective dynamics whenever a swarms’ expansion is arrested.

The data presented could be explained by a different hypothesis rather than a MIPS transition. For example, just the fact that biofilm formation requires high cell density. The experiments are carried out on biofilm inducing medium – this suggests that only high cell density is needed for biofilm formation and if the authors waited long enough biofilms would form under all the conditions tested (irrespective of antibiotic application or swarming). There is not a great deal of evidence that the MIPS transition is actually playing a role. For example, the authors use growth on hard agar (which does not support swarming) as evidence that swarming is required for biofilm formation. But the hard agar plates are made with biofilm promoting media, so presumably the only reason no biofilms are seen is that the cell density is not high enough. In the hard agar control, at the beginning of the experiment cells are spread across the whole plate.What happens with an inoculum started in the centre of the hard agar plate (similar to the swarming experiments) and a KAN disc placed nearby. Too many differences exist between the hard agar and swarming experiments (e.g. number of cells, placement of antibiotic disk in different location on dish, method for inoculum placement) to be able to say clearly that swarming is required for biofilm formation near the KAN gradient.

We believe that this issue comes from a misunderstanding of the message that we are trying to convey, which we agree can probably be explained better. We agree with the reviewer that high cell density is essential for biofilm formation. For example, a previous study by one of the authors (Asally et al., 2012) showed that creating local high-cell density regions enables prescribed patterning of biofilm wrinkles. Our point here is that high cell density can be achieved by different mechanisms. For *B. subtilis* swarms, which expand as a monolayer, arresting the swarm progression leads to a localised accumulation of cells at the front. We believe that it is this localised accumulation which then promotes wrinkle formation. MIPS here enters as the spontaneous mechanism leading to the initial accumulation. Of course, cell density can be increased in many different ways. However, for swarming *B. subtilis*, when the front progression is arrested this happens spontaneously through a MIPS-like mechanism. We would like to stress that, a priori, it did not have to be like this. The swarming monolayer could have just stopped and remained a monolayer until all cells became immotile. We believe that in this case the system would *not* have produced the wrinkles we observe, and in fact wrinkles do not form far from the kanamycin disk (or from boundaries), where the *localised* accumulation we describe does not happen. Altogether our observations imply that, for a *B. subtilis* swarm, the observed local emergence of wrinkles is the consequence of the local cell accumulation caused by the arrest of the swarm’s front. This cell accumulation happens through a MIPS-like transition.

Regarding the question of whether biofilm would have developed anyway in control swarming plates had the experiments run for long enough time, we have re-run swarming plates without kanamycin and kept them for 7 days. Figure 2—figure supplement 5 shows a representative example, and we can see that no wrinkles developed far from the plate’s boundary (for the region close to the plate boundary, see the Results and Author response image 1). This behaviour is representative of all the 10 replicates.

We believe that waiting 7 days is appropriate, since it has been shown that wrinkle biofilms complete developing in 1.5 days and begin to relax after 3 days (Leiman et al., 2013). Consistent with this, we observed that the wrinkled morphology in the kanamycin and barrier experiments is most prominent after 2 days post inoculation, and then slowly decayed over time.

Regarding the control with hard agar, we originally spread cells evenly across the plate to mimic the uniform distribution in our swarming assay. We have now repeated this experiment with the kanamycin disk next to the edge of the petri dish to make the control in hard agar more similar to the swarming experiments, as suggested by the reviewer. This has now been included in the amended Figure 3—figure supplement 2. The figure shows that there does not appear to be any noticeable difference between the weak wrinkles that form at the boundary around the kanamycin disk when this is placed either at the centre or at the boundary of the petri dish. We note that these wrinkles might well be appearing due to the presence of motile bacteria within in the culture that we spread on the plate, although at present we have not tested this.

To further address the issues raised by the reviewer, we have decided to look at the effect of spotting a colony on hard agar at different distances from the kanamycin disk. We note that when a colony is spotted on hard agar, it does not expand more than ~1 cm (Figure 2—figure supplement 6). On a kanamycin hard agar pad, made as described in the main paper, we spotted 4 *μ* l of culture at 4 different positions (4 *μ* l at each), as shown in Figure 2—figure supplement 6. The closest, ~2 cm away from the kanamycin disk, shows a strong inhibition of wrinkle formation. The two on the sides, 2 cm away from the previous one, show an inhibition of wrinkle formation on the side of the inoculum towards the kanamycin disk. These should be compared to the normal growth that can be observed in the farthest colony. Overall, we see that kanamycin inhibits wrinkle formation. This is consistent with our previous reports of lack of strong wrinkles on hard agar around the kanamycin disk.

The double antibiotic experiment appears to have an issue. As I understand it the design of the experiment is as follows “We administer a total amount of 200 µg of kanamycin in two steps, an initial one when placing the disk on the plate, and the second as the swarming front stopped (Figure 5B, Figure 6—figure supplement 2). The emergence of wrinkles was greatly suppressed when kanamycin was administrated sequentially, despite keeping the total amount of antibiotic constant (Figure 5B, C and Figure 6—figure supplement 2).”I am concerned that the authors do not know exactly how much antibiotic the cells are experiencing as the antibiotics will diffuse over time into the plate (as the authors show in Figure 2—figure supplement 1). Are the authors sure that the cells at the edge of the disc are really experiencing the same amount of antibiotic if the antibiotic is added 24 h later? I would assume that the concentrations would be higher for the cells at the swarming front on the second application due to less time for the kanamycin to diffuse (as suggested by Figure 2—figure supplement 1). Could this explain why the second application is so effective? Apologies if this is controlled for and I missed it.

We agree with the reviewer that this is an important point, and we would like to address this in two steps.

Firstly, we would like to stress that it was not our aim to guarantee that the cells experience always the same amount of antibiotics. We wanted to *use* the same amount, but that does not per se guarantee that the cells are exposed to the same amount (we agree with the reviewer on this, but see below). The question that we wanted to pose is whether, without increasing the human use of antibiotics, there are antibiotic-administration strategies that are better at preventing -or at least reducing- the emergence of the wrinkly biofilm we observe. According to our experiments, the answer to this question is yes. Splitting the same total amount of antibiotic added to the disk into a low initial dose followed by a large one, brings a noticeable reduction in the wrinkle wavelengths compared to using all of the antibiotic in a single initial dose. This translates into a weaker biofilm (see also our reply to comment #3 from reviewer #1). To clarify this point, we amended the sentence “The emergence of wrinkles was greatly suppressed when kanamycin was administrated sequentially, despite keeping the total amount of antibiotic constant (Figure 6B, C and Figure 6—figure supplement 2).” to read “The emergence of wrinkles was greatly suppressed when kanamycin was administrated sequentially, despite keeping constant the total amount of antibiotic added to the disk (Figure 6B, C and Figure 6—figure supplement 2).”

At the same time, following the reviewer’s advice, we tried to measure the local antibiotic concentration at the distance from the kanamycin disk where the wrinkles are formed. To this end, we used a Kirby-Bauer assay to measure the antibiotic concentration profile at 5 different positions along the line connecting the centre of the petri dish and the disk (see Figure 6—figure supplement 1). We performed the measurements at 0h, 20h, 44h, starting when the front propagation towards the kanamycin disk stopped. Each time point measurement was repeated 3 times for each of the administration protocols. Different plates were used at different time points for a total of 36 plates. We stress that the experiments were performed on inoculated swarming plates, not on blank plates with just the kanamycin disk. This is because the spread of the antibiotic could in principle be influenced by surface effects (e.g. Marangoni flows driven by biosurfactants) that would be different between a plate with a swarming colony and one without. The curves were then used to estimate the antibiotic concentration in the area where the wrinkles form, which is dependent on the antibiotic dosage protocol. The results can be seen in Figure 6A. As shown in Figure 6A, the estimated kanamycin concentration at the wrinkle location is compatible across the different dosage protocols with a potential minor difference for the 150-µg case. The discrepancy observed in this case is possibly the result of non-diffusive transport.